# Recursive Structure Discovery as an Inductive Bias for Symbolic Regression

## Abstract

Symbolic regression (SR) can recover analytic laws from data, but its search space is enormous. Many scientific targets are structurally simple, for example additively or multiplicatively separable, yet most SR pipelines do not exploit this. We introduce a recursive structure discovery step that tests for separability using accurate derivatives from a small neural model trained with second-order updates. The method decomposes $y = f(\mathbf{x})$ into a hierarchy of simpler subfunctions, which we feed to SR as a structure prior. This plug-in reduces search complexity, improves interpretability, and can attach to any SR backend; here we pair it with a deep RL generator. This substantially reduces search complexity, improves interpretability, and remains robust to noise, maintaining reliable separability detection under challenging conditions. On SRBench (Feynman, 120 equations), the structure-aware pipeline achieves state-of-the-art exact recovery, outperforming separability-only, pure RL, and prior hybrid baselines.

## 1 Introduction

Many laws in the natural sciences exhibit modular, decomposable structure. Historical examples make this concrete: Kepler's analysis recast astronomical observations as elliptical orbits; the ideal gas law isolates pressure, volume, and temperature in a separable relation; Lotka–Volterra models express population dynamics through interacting components. These examples suggest that useful scientific models often emerge by uncovering how variables combine through simple, interpretable structure. Yet in practice, we rarely observe these components directly. Instead, we measure aggregates signals produced by multiple interacting processes. The scientific task is to reverse-engineer this composition: to infer the hidden modular structure that gives rise to the observations. While deep learning models excel at fitting such data, they rarely expose this kind of structure. Their internal representations are not naturally separable or composable in the way scientific laws tend to be. The challenge, then, is to develop methods that combine the flexibility of modern learning with the interpretability of modular decompositions.

This paper addresses that gap by focusing on *structure discovery* as a precursor to symbolic regression (SR). Our aim is not simply to regress an unknown function directly, but to automatically uncover its latent building blocks (additive or multiplicative separability, and compositionality through simple transformations) that have long underpinned interpretable physical models. In classical SR, the search over functional forms is combinatorial and hard. Practitioners typically mitigate this with expert-provided inductive biases about how variables interact. To make this injection of prior knowledge scalable, we propose using neural networks not as final predictors but as instruments to expose structural patterns (separability and simple compositions) and pass these constraints to SR. The discovered hierarchy then constrains the SR search space, guiding it toward candidate expressions that are both accurate and human-readable. In this way, our approach integrates the predictive power of modern machine learning with the interpretability of symbolic modeling, making SR more tractable while staying faithful to the organizing principles of scientific equations.

**Unveiling graph structure in data** To address this challenge, we propose a neural network framework that explicitly discovers hierarchical structures in science datasets. Given a dataset $\{\mathbf{x}, y\}$ and an underlying function $f$ such that $y = f(\mathbf{x})$, our approach identifies two fundamental decomposition patterns: (1) additive/multiplicative separability – e.g., that the target can be modeled as $y = f_1(\mathbf{x}_1) + f_2(\mathbf{x}_2)$ for subsets $\mathbf{x}_1, \mathbf{x}_2 \subset \mathbf{x}$ and (2) constitutionality through nonlinear unary functions

| | | **Ours** | uDSR | PhySO | AIF | uDSR-A | DSR |
|---|---|---|---|---|---|---|---|
| | RL-based SR | ✓ | ✓ | ✓ | | ✓ | ✓ |
| | Large Scale Pre-training | | ✓ | | | | |
| | Genetic Programming | | ✓ | | | | |
| | Dimensional Analysis | ✓ | | ✓ | ✓ | | |
| Derivative- | Additive Separability | ✓ | ✓ | | ✓ | ✓ | |
| based | Multiplicative Separability | ✓ | | | | | |
| Structure | Nested Separability Detection | ✓ | | | | | |
| Analysis | Structural Prior Guidance | ✓ | | | | | |
| | Feynman Benchmark Recovery Score | **72 %** | 69 % | 58 % | 55 % | 50 % | 43 % |

Table 1: **Overview of structural analysis features supported by our method compared to existing SR frameworks.** Our approach uniquely combines derivative-based separability detection, nested decomposition, dimensional analysis, and a structural prior, leading to the highest score on the Feynman benchmark. uDSR-A denotes an ablation of uDSR containing only the DSR and AIF components. Baseline results from (La Cava et al., 2021; Landajuela et al., 2022; Tenachi et al., 2023)

(e.g., $\frac{1}{\square}, \square^2, \sqrt{\square}, \exp, \log, \cos \ldots$) – e.g. $y = \sqrt{f_1(\mathbf{x_1}) + f_2(\mathbf{x_2})}$. These patterns form the natural building blocks of analytical physical representations. Given a function $f : \mathbb{R}^{n_x} \to \mathbb{R}$, we decompose it into a tree of sub-models $\{f_i\}_{i=1}^N$, each operating on distinct input subsets. We then represents the target function as $f = \Phi \circ (f_1, \ldots, f_N)$, where the composition function $\Phi$ hierarchically combines sub-models through either elementary operations $(+, \times)$ or nonlinear transformations. The resulting tree – composed of neural sub-models and symbolic functions – is already more interpretable than a monolithic network, yet its interpretability can be further enhanced.

**Symbolic Regression (SR)** We use the extracted hierarchical structure to directly inform a symbolic regression (SR) process, which infers an analytical form for $f$ from data $(\mathbf{x}, y)$. The goal here is to find a sequence of mathematical operators (e.g., $+, \times, \sin, \exp \ldots$), input variables and free constants such that the given function is approximated as well as possible. This combinatorial search is NP-hard (Virgolin & Pissis, 2022), making prior knowledge of the data's structure (eg. separability which is critical for efficiency). Our work bridges this gap by automatically detecting and exploiting such structure.

In §1.1, we contextualize our contributions within existing literature on SR and structural decomposition.

## 1.1 RELATED WORKS & CONTRIBUTION

**Traditional SR Methods** SR has historically relied on genetic programming (GP), emulating natural evolution to explore equation spaces. This approach underpins frameworks like `Eureqa` (Schmidt & Lipson, 2009), `PySR` (Cranmer, 2023), and others (Stephens, 2015; Cava et al., 2019; Kommenda et al., 2020; Virgolin et al., 2021).

**Deep Learning-Based SR** Recent advances employ neural networks for SR through two dominant approaches: (1) large-scale pre-training of generative transformer models to map datasets to corresponding equations (Kamienny et al., 2022; Lalande et al., 2023; Biggio et al., 2020; 2021), and (2) deep reinforcement learning (RL) where networks iteratively generate and refine equations via policy gradient methods, as in `DSR` (Petersen et al., 2021; Landajuela et al., 2021) and `PhySO` (Tenachi et al., 2023; 2024). Our work extends the second paradigm, utilizing its inherent flexibility to incorporate structural priors as an inductive bias and per-submodel length constraints.

**Neuro-Symbolic Approaches** Alternative approaches embed symbolic operations (e.g., $\frac{1}{\square}, \square^2, \exp$ ...) within compact neural architectures, enforcing sparsity to recover interpretable equations (Fiorini et al., 2024; Scholl et al., 2023; Martius & Lampert, 2017; Brunton et al., 2016; Sahoo et al., 2018). While similarly incorporate nonlinearities, our approach differs by first hierarchically composing sub-models without explicit sparsity constraints, and only then applying symbolic regression to them.

**Separability Detection** Prior work on separability-leveraging SR includes `AIF` (Udrescu & Tegmark, 2020; Udrescu et al., 2020) and its RL hybrid `uDSR` (Landajuela et al., 2022)[1]. These approaches

---

[1] In addition to combining RL-based SR and `AIF`, `uDSR` also integrates large-scale pre-training and a genetic programming approach.

are primarily focused on detecting additive separability and are limited in handling the full range of structures, specifically:

1. They cannot detect separabilities nested in nonlinear functions (e.g., $f(\mathbf{x}) = g(f_1(\mathbf{x}_1) + f_2(\mathbf{x}_2))$ where $g$ is a nonlinearity).

2. Although these approaches employ a heuristic scheme for multiplicative separability, this method lacks a robust foundation in derivative properties and consequently yields poor performance (the method and its limitations are documented in Appendix D)..

Our method addresses these limitations by introducing a robust, derivative-based detection scheme for both additive and multiplicative separability. This rigorous foundation, coupled with the ability to test for separability under common non-linear transformations, enables a fully recursive decomposition of complex functions into a reliable hierarchical structure.

**Gradient Estimation** Accurate separability detection requires precise gradient estimation. While derivative-constrained training is well-studied in physics-informed neural networks (PINNs) (Raissi et al., 2019) and Sobolev training (Czarnecki et al., 2017), few works address derivative estimation from data alone. We address this by employing `NestyNet`, a compact neural architecture designed for high-fidelity derivative computation from potentially noisy observations.

We summarize shared components between our method and closely related SR frameworks, alongside our key contributions, in Table 1. Sections 2–4 detail our methodology, results, and conclusions respectively.

## 2 METHOD

As shown on Figure 1, our framework operates in two main stages:

(a) We first model a dataset $\{\mathbf{x}, y\}$ by recursively detecting separabilities in the input variables and organizing them into a tree of compact neural networks. This produces a neuro-symbolic structure that is more interpretable than a monolithic network, though no symbolic regression is applied at this stage making it compatible with any SR algorithm.

(b) The resulting structure—initially composed of black-box subnetworks—is then converted into a symbolic expression through RL-based SR, which leverages the discovered hierarchy as a structural prior to guide the search.

In practice, we employ `NestyNet` as a high-fidelity function emulator (Section 2.1), taking advantage of its precise derivative estimates to recursively detect additive and multiplicative separabilities, including those nested within nonlinear functions (Section 2.2). The resulting interpretable graph of sub-functions informs a deep RL-based SR process through structural priors (Section 2.3), focusing the search on analytically plausible expressions consistent with the discovered hierarchy.

### 2.1 DERIVATIVES EMULATOR

**Learning Precise Derivatives** To detect separabilities, we require an emulator $f$ that not only fits the data $(\mathbf{x}, y) \in \mathbb{R}^{n_x \times n_y}$ accurately but also yields reliable first- and second-order derivatives, $\partial f / \partial \mathbf{x}$ and $\partial^2 f / \partial \mathbf{x}^2$. For this purpose, we employ `NestyNet`[2]—a shallow neural module which's instances can be composed in a tree structure to emulate functions and their derivatives – the key component enabling derivative-based separability detection. Its strong regularization comes from an extremely compact parameterization (around $\sim 100$ parameters), which reduces overfitting. This compactness is enabled by the Levenberg–Marquardt (LM) optimizer (Levenberg, 1944; Marquardt, 1963), which leverages full Jacobian propagation—computing derivatives for each data point with respect to (w.r.t.) input variables or model parameters—throughout the architecture, rather than relying solely on gradients. In this work, we restrict attention to problems with arbitrary input dimension $n_x$, but scalar output $n_y = 1$.

---

[2] `NestyNet` will be fully described in a forthcoming paper by one of the anonymous co-authors. In this work, we focus only on its role as a tool for recursive structure discovery, specifically its ability to operate within a composite tree of sub-models while providing accurate derivative estimates.

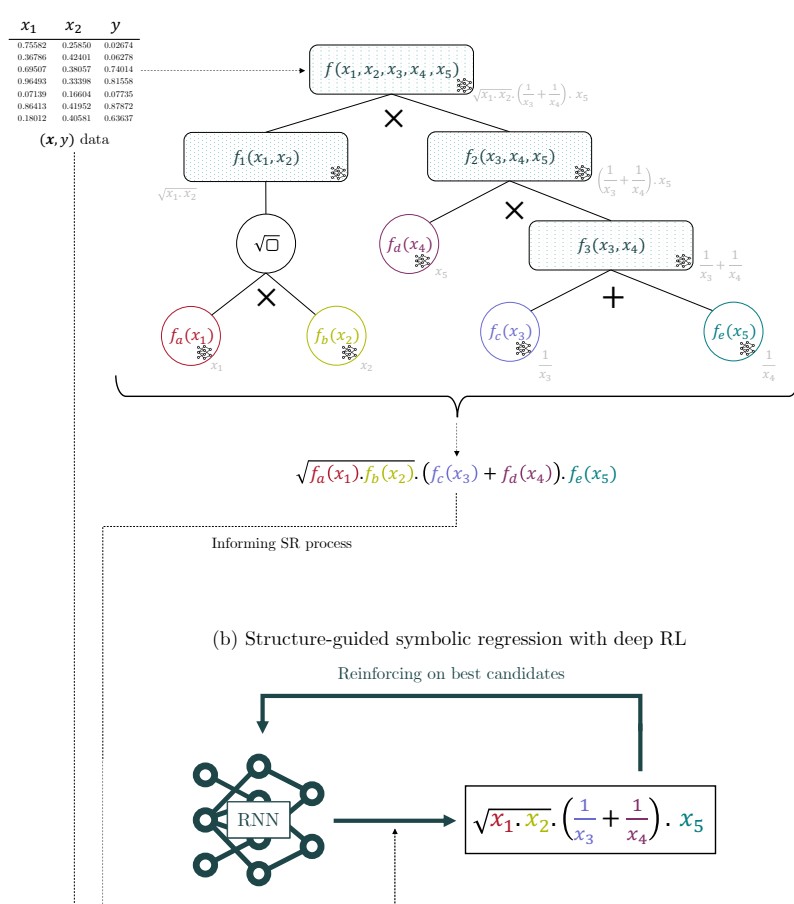

Figure 1: **Structure-Aware Symbolic Regression.** (a) Automated decomposition of an intricate dataset $(\mathbf{x}, y)$ into an interpretable tree structure composed of neural models through recursive detection of additive/multiplicative separabilities, including those nested within nonlinear operations. (b) Structure-informed inference where the discovered hierarchy guides SR through a prior, enabling exact recovery of the ground-truth equation via deep RL.

**NestyNet Layer** A single `NestyNet` layer is defined as:

$$y = a \, \log\big(1 + \exp(K\mathbf{x} + b)\big), \tag{1}$$

where $K \in \mathbb{R}^{h \times n_x}$ is a weight matrix, $b \in \mathbb{R}^h$ a bias vector, and $a \in \mathbb{R}^{n_y \times h}$ a scaling matrix. The set of trainable parameters is $\theta = \{K, b, a\} = \{\theta_i\}_{i < n_{\text{params}}}$.

This parametrization admits closed-form expressions for both the Jacobian and Hessian w.r.t. the inputs. Denoting by $\sigma(\cdot)$ the logistic sigmoid and by $\sigma'(\cdot) = \sigma(\cdot)\,(1 - \sigma(\cdot))$ its derivative:

$$\nabla_{\mathbf{x}} y = K^\top \big(a \, \sigma(K\mathbf{x} + b)\big), \tag{2}$$

$$\frac{\partial^2 y}{\partial x_i \, \partial x_j} = K^\top \big(a \, \sigma'(K\mathbf{x} + b)\big) K. \tag{3}$$

In addition, derivatives w.r.t. the model parameters $\{K, b, a\}$ (given in Appendix A) can also trivially be written in closed form which enables LM optimization.

**Levenberg-Marquardt (LM) Optimization** While the universal approximation theorem (Hornik et al., 1989) guarantees that this shallow architecture can represent any smooth function given suffi-

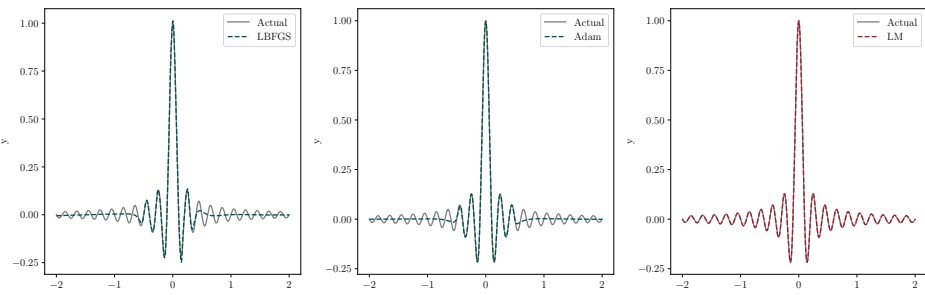

Figure 2: **Second-order optimization advantage on compact parametrizations.** Training dynamics of a `NestyNet` layer with 272 parameters under different optimization strategies. The Levenberg–Marquardt (LM) method outperforms both `Adam` (first-order) and `L-BFGS` (Zhu et al., 1997) (quasi-Newton) in convergence speed and final accuracy, especially on subtle features.

cient width $h$, traditional gradient descent approaches—which rely on minimizing a scalar loss function $\mathcal{L}(\theta) = \sum\limits_{k<n_{\text{samples}}} (y^{\langle k \rangle} - f(x^{\langle k \rangle}, \theta))^2$ via partial derivatives $\nabla_\theta \mathcal{L} = (\partial \mathcal{L}/\partial \theta_1, \ldots, \partial \mathcal{L}/\partial \theta_M)$—prove impractical for complex tasks with optimizers like `Adam` (Kingma & Ba, 2015) or `SGD` (Robbins & Monro, 1951). This limitation motivated the development of deep architectures (LeCun et al., 2015). We instead employ the LM algorithm. The method computes the Jacobian matrix $J_{kj} = \partial f(x^{\langle k \rangle})/\partial \theta_j$ (across samples : $k < n_{\text{samples}}$ and model parameters : $j < n_{\text{params}}$) and approximates the explicit Hessian as $J^\top J$, then solves the linear system $(J^\top J + \lambda I)\Delta\theta = J^\top(y - f(x, \theta))$ for parameter updates. Unlike first-order methods (e.g., SGD) that only estimate descent directions, this approach directly computes parameter updates $\Delta\theta$ to minimize $\chi^2$ by combining gradient and curvature information.

As shown in Figure 2, this second-order optimization captures fine-scale data variations more accurately, even for compact models (Ranganathan, 2004). Furthermore, Figure 3 illustrates that, due to its strong regularization via an extremely compact parameterization enabled by LM optimization, the `NestyNet` yields significantly more accurate derivatives w.r.t. input variables than a standard multilayer perceptron.

**Composite Model** Our goal is to detect and model separabilities in data. To this end, we construct a tree composed of multiple `NestyNet` layers $\{f_i\}_{i=1}^N$, each operating on distinct subsets of the input. The target function is modeled as : $f = \Phi \circ (f_1, \ldots, f_N)$. Where the composition function $\Phi$ hierarchically combines submodels through either elementary operations $(+, \times)$ or nonlinear transformations. This structure is illustrated in Panel a of Figure 1. In practice, the composite model is initialized with a single `NestyNet` layer acting on all input variables. If a separability is detected, this layer is replaced by two new layers operating on the corresponding variable subsets, joined by the appropriate composition operator: addition for additive separability, multiplication for multiplicative separability, or a nonlinear wrapper when the separability is nested inside a transformation. This procedure is then applied recursively to each new sub-layer until no further separabilities are found, yielding a full tree structure.

To evaluate and train such a tree using the LM optimizer, we must propagate several key differential quantities. Consider any node $f_v \in \{f_i\}_{i=1}^N$. If $f$ has two children, $f_{v_1}$ and $f_{v_2}$, combined via a binary operation $f_v = g(f_{v_1}, f_{v_2})$, $g$ can be either addition $(f_v = f_{v_1} + f_{v_2})$ or multiplication $(f_v = f_{v_1}.f_{v_2})$. If $f_v$ has a single child, $f_{v_1}$, it is of the form $f_v = g(f_{v_1})$, where $g$ is a nonlinear transformation. For each node, we propagate (i) full Jacobians and Hessians w.r.t. inputs, (ii) Jacobians w.r.t. model parameters, (iii) Jacobian- and vector–Jacobian products, and (iv) the diagonal of $J^\top J$ w.r.t. parameters; closed-form propagation rules for unary operations such as $g \in \{+, \times, \square^{-1}, \square^2, \sqrt{\square}, \exp(\square), \log(\square), \cos(\square)\}$ are given in Appendix A.

**Fitting** Details of the `NestyNet` composite fitting procedure and hyper-parameters are given in Appendix B.


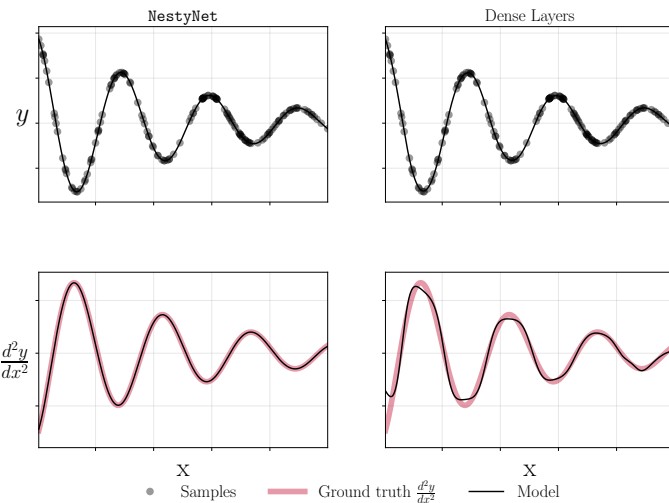

Figure 3: **NestyNet vs. dense layers: derivative accuracy.** Comparison of fit quality and input derivatives between the compact `NestyNet` (272 parameters) trained with LM and a standard dense network (3811 parameters). For fairness, each model was configured with the minimal number of parameters needed to reach test set $R^2 > 0.999$.

## 2.2 SEPARABILITY DETECTION

Given a dataset $(\mathbf{x}, y)$, we first train an instance $f$ of the regularized `NestyNet` layer (described above) to model the data. This provides access to high-fidelity derivatives w.r.t. the input variables $\mathbf{x}$ which we examine in inference mode.

**Additive separability** We detect additive separability by testing whether our emulator $f$ decomposes into sub-functions $f_1$ and $f_2$ operating on different input subsets $\mathbf{x}_1$ and $\mathbf{x}_2$, satisfying $y = f_1(\mathbf{x}_1) + f_2(\mathbf{x}_2)$. In this case, the mixed second-order partial derivatives vanish, i.e., $\frac{\partial^2 y}{\partial x_i \partial x_j} = 0$ for $i \neq j$. To verify separability between a variable pair $(x_i, x_j)$, we therefore evaluate the condition:

$$\text{med}\left(\left|\frac{\partial^2 y}{\partial x_i \partial x_j}\right|\right) < \epsilon_{\text{add}} \tag{4}$$

Where $\epsilon_{\text{add}}$ is an empirically determined threshold for negligible interactions (with med denoting the median operation across sample points). This test is applied across variable pairs and input partitions, when multiple valid separations exist, we select the configuration minimizing $\dim(\mathbf{x}_2)$ to prioritize the most interpretable decomposition.

**Multiplicative Separability** Similarly, we detect multiplicative separability by testing whether our emulator $f$ decomposes into sub-functions $f_1$ and $f_2$ operating on distinct input subsets $\mathbf{x}_1$ and $\mathbf{x}_2$, such that $y = f_1(\mathbf{x}_1) \cdot f_2(\mathbf{x}_2)$. To verify separability between a pair of variables $(x_i, x_j)$ where $i \neq j$, we test for the form $y = f_1(x_i) \cdot f_2(x_j)$, allowing for a potential additive scalar constant $\beta$ giving $y = f_1(x_i) \cdot f_2(x_j) + \beta$. This is done by verifying the mixed second-order partial derivatives relationship:

$$\text{med}\left(\left|\frac{\partial^2 y}{\partial x_i \partial x_j}\frac{1}{(y - \beta_{\text{med}})} - \frac{\partial y}{\partial x_i}\frac{\partial y}{\partial x_j}\frac{1}{(y - \beta_{\text{med}})^2}\right|\right) < \epsilon_{\text{mul}} \tag{5}$$

where $\epsilon_{\text{mul}}$ is an empirically determined threshold for negligible interactions, and $\beta_{\text{med}} = \text{med}(\beta)$, with: $\beta = y - \frac{\partial y}{\partial x_i} \cdot \frac{\partial y}{\partial x_j} \cdot \frac{1}{\partial^2 y/\partial x_i \partial x_j}$. Since $\beta$ should be constant across all input samples when the separation is valid, we additionally verify that its scaled scatter is small:

$$\frac{\text{med}(|\beta - \text{med}(\beta)|)}{\text{med}(|y|)} < \epsilon_{\beta_{mad}} \tag{6}$$

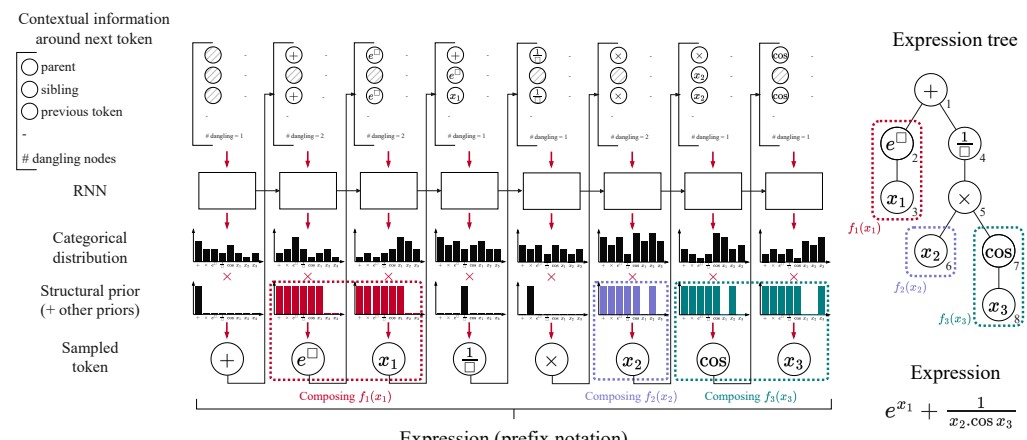

Figure 4: **RNN-based expression generation with a structural prior.** An RNN generates a symbolic expression sequentially in prefix notation. At each step, it outputs a categorical distribution over tokens, modulated by a deterministic structural prior derived from a discovered separability tree. The adjusted distribution is sampled to select the next token, and this process is repeated until the full expression is obtained, which can be represented as a tree or in human-readable form. Here, the structural prior encourages a form such as $f_1(x_1) + \frac{1}{f_2(x_2) \cdot f_3(x_3)}$, while allowing the RNN to freely compose the sub-functions $f_1$, $f_2$, and $f_3$. For example, starting with the $+$ operator in prefix notation enforces separation between the branch depending on $x_1$ and the branch depending on $(x_2, x_3)$, while forbidding variables outside each branch during sub-function generation.

Where $\epsilon_{\beta_{mad}}$ is an empirically determined threshold for negligible scatter. Again, when multiple valid separations exist, we select the configuration minimizing $\dim(\mathbf{x}_2)$ to prioritize the most interpretable decomposition. Numerical values for threshold parameters are given in Table 2 and the corresponding criteria used in noisy scenarios are detailed in Appendix E.

| Criterion | Threshold Value |
|---|---|
| $\epsilon_{\text{add}}$ | $10^{-4}$ |
| $\epsilon_{\text{mul}}$ | $10^{-12}$ |
| $\epsilon_{\beta_{\text{mad}}}$ | $10^{-3}$ |

Table 2: **Detection thresholds for separability analysis**

**Recursive Structure Search** Upon detecting separability, we emulate each sub-function using distinct NestyNet instances and fit the resulting composite tree of sub-models. Separability tests are then performed on tree nodes where necessary—that is, on multivariate nodes where tests have not yet been applied. This modular, recursive approach enables iterative decomposition, allowing each sub-function to undergo further separability analysis and progressively transforming complex datasets into interpretable graphs of simpler models.

When no separability is detected in $f(\mathbf{x})$, we additionally examine separabilities on transformed versions $g_{\text{NL}}(f(\mathbf{x}))$ using a library of common nonlinearities: $g \in F_{\text{NL}}$ with $F_{\text{NL}} = \{\Box^{-1}, \Box^2, \sqrt{\Box}, \exp, \log, \cos\}$. For this step, we re-fit a composite tree of NestyNet instances that incorporates the trial nonlinear transformation before the node of being tested for separability. This capability allows the discovery of nested structures such as $f(x_1, x_2) = \sqrt{f_1(x_1) + f_2(x_2)}$, as illustrated in Figure 1 (Panel a).

## 2.3 SYMBOLIC REGRESSION (SR)

**Structural Prior Integration** The discovered graph structure guides the SR process through probabilistic priors, improving search efficiency without imposing rigid constraints. Our framework

uses a recurrent neural network (RNN) to sequentially generate trial expressions in prefix notation[3], while dynamically encouraging structural compatibility. At each step, the RNN outputs a categorical distribution over the token vocabulary, from which a token is sampled. Repeating this process generates a complete expression in prefix form, which can be straightforwardly converted back to standard human-readable notation (i.e. infix notation). Throughout generation, the algorithm tracks its position within the separability tree and deterministically adjusts the RNN's output distribution according through a 'structural' prior, ensuring compliance with the constraints of each node.

Specifically, when composing a sub-expression associated with a separated node, the probability of sampling input variables outside the allowed subset is reduced to near-zero values[4]. At separability nodes, the distribution is biased toward relevant operators: additive separabilities favor $\{+, -\}$, multiplicative separabilities favor $\{\times, /\}$, and nodes representing nonlinearities from $F_{\text{NL}}$ enforce the corresponding operator. This approach preserves the RNN's exploratory capability while strongly encouraging structurally valid expressions. This process is illustrated on Figure 4.

We complement these constraints with length priors on sub-functions, initially based on the number of input variables as a proxy for complexity. Specifically, each sub-expression is encouraged to have a length roughly equal to $2 \times$ (number of input variables), using a Gaussian soft prior with scale 1.5. More sophisticated complexity measures can be incorporated in future implementations.

**Reinforcement learning-based SR** We employ a deep reinforcement learning (RL) approach to train the expression generator policy. At each iteration, a batch of expressions is generated according to the structural prior described above, as well as additional priors. Each generated expression is evaluated based on its fit quality to the dataset $\{\mathbf{x}, y\}$, first optimizing any free constants using `LBFGS` (Zhu et al., 1997), leveraging automatic differentiation of the trial expressions.

For a trial expression $f$, we define the reward as $R = 1/(1 + \text{NRMSE})$ where NRMSE denotes the normalized root mean squared error. $R \in [0, 1]$ where 1 denotes a perfect fit. Policy gradients are then approximated based on the top 5% of candidates, following the risk-seeking optimization strategy (Petersen et al., 2021) adapted from (Rajeswaran et al., 2017). Training continues until the policy consistently proposes expressions that fit the data well. Further details on our RL-based symbolic regression strategy are provided in Appendix C.

The complete workflow (Figure 1, Panel b) thus combines structural awareness with the expressive power of deep learning SR, where priors focus the search space.

## 3 EXPERIMENTS

**Separability Benchmark** We introduce a separability benchmark designed to evaluate the performance of our separability-detection algorithm, i.e., the pre-processing stage preceding symbolic regression. The benchmark comprises 24 synthetic datasets for which the ground-truth separability structure is known. These include additively separable cases, multiplicatively separable cases, instances where separability is embedded within non-linear transformations, and critically, cases that contain *no* separability—allowing us to assess the false-positive rate. Full descriptions of all 24 challenges and evaluation protocol are provided in Appendix E.

A summary of results is presented in Table 3. Our method reliably detects both additive and multiplicative separabilities, including cases where they are nested within non-linear functions. It also exhibits strong robustness to noise, retaining most of its detection capability even at a 10% noise level. Importantly, even though higher noise can obscure true separabilities, the method remains conservative: across all tests and noise levels, it produces *no false positives*.

**Feynman Benchmark** We now evaluate the full pipeline—separability detection followed by structure-aware symbolic regression. We benchmark our method using the standardized `SRBench` framework ⟡ cavalab/srbench (La Cava et al., 2021), comparing against 18 baseline methods on 120 ground-truth equations from the Feynman SR benchmark (Udrescu & Tegmark, 2020) to be exactly recovered from their associated data. The full benchmarking protocol is given in F.

---

[3]Prefix notation writes operations before their arguments, eliminating the need for parentheses. For example, $a + \cos(b)$ is written as $\{+, a, \cos, b\}$ in this notation. This notation can be obtained by representing an expression as a tree and listing its nodes first in depth and then left to right as illustrated on Figure 4

[4]Kept non-zero to preserve theoretical exploration capacity.

| Separability | | Noise | | | |
|---|---|---|---|---|---|
| | | 0 % | 1 % | 5 % | 10 % |
| Direct separability | $f_a + f_b$ | 100.0 | 81.2 | 75.0 | 75.0 |
| | $f_a \cdot f_b$ | 100.0 | 97.7 | 86.4 | 72.7 |
| Nested separability | $g(f_a + f_b)$ | 100.0 | 90.0 | 80.0 | 60.0 |
| | $g(f_a \cdot f_b)$ | 100.0 | 100.0 | 50.0 | 75.0 |
| No separability | | 100.0 | 100.0 | 100.0 | 100.0 |
| | | **100.0** | **93.8** | **78.3** | **76.5** |

Table 3: **Strong robustness to noise and false positives.** Separability-detection accuracy (in %) across 24 challenges, evaluated at multiple noise levels. Cases include additive, multiplicative, and nested separabilities, as well as non-separable datasets. The method maintains high accuracy under noise and produces no false-positive detections even at higher noise levels.

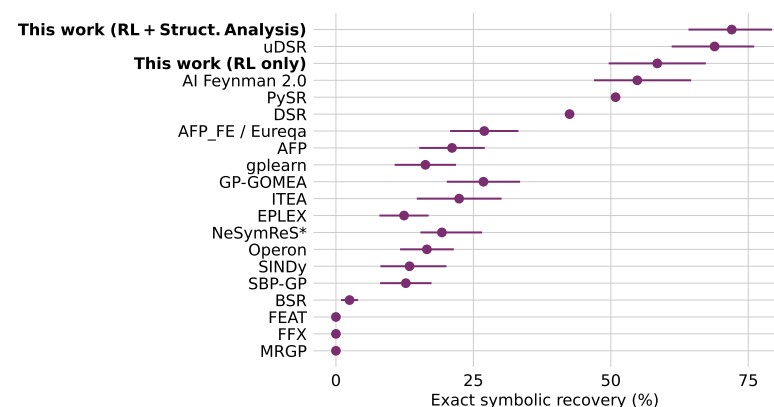

Figure 5: **State-of-the-Art Performance on SRBench.** Exact symbolic recovery rates are reported for 120 Feynman problems from `SRBench` (La Cava et al., 2021). Our method outperforms all baselines including traditional structure analysis methods, pure deep RL methods, and previous hybrid approaches. The plot also includes an ablation where our RL-based SR is applied without prior structure analysis. When available, error bars represent 95% confidence intervals. Baseline results from (La Cava et al., 2021; Landajuela et al., 2022; Grayeli et al., 2024)

Figure 5 reports our method's performance relative to baseline approaches. We additionally include an ablation where the pre-processing structure analysis is omitted and RL-based symbolic regression is applied directly. Our method (RL with structure analysis) achieves state-of-the-art exact recovery (72%), outperforming: (1) `AIF` (the original separability-based approach), (2) pure RL methods (`DSR`, our RL-only ablation), and (3) hybrid approaches (`uDSR`).

This advancement stems from three key innovations: (i) handling multiplicative separabilities, (ii) detecting separabilities nested within nonlinear transformations, and (iii) effectively integrating structural inference with RL through adaptive priors rather than rigid decomposition.

## 4 DISCUSSION & CONCLUSION

**The Value of Symbolic Constraints** It is essential to note that many SR exploration strategies are driven by accuracy with minimal constraints on symbolic arrangement. However, the optimization paths that maximize fit quality and those that lead to exact symbolic recovery are not necessarily aligned. In practice, a model can achieve increasingly accurate fits while simultaneously diverging from the correct underlying expression. By introducing structural information via separability-based analysis, we provide guidance on symbolic arrangements, steering the search toward more faithful solutions. Importantly, rather than enforcing a rigid constraint, our approach implements this guidance as a *soft* inductive bias through a probabilistic prior. This is particularly valuable for RL and genetic programming based SR methods, which traditionally access the data only through a scalar measure

of fit quality. Supplementing this with structural information about symbolic arrangements therefore provides a crucial additional signal.

**Robustness** Our separability detection is highly robust to noise and, even more importantly, extremely conservative with respect to false positives. While high noise can occasionally mask true separabilities, the method never reports separability when none is present—standing in clear contrast with previous separability techniques. Two core innovations underlie this resilience: (1) the stability of `NestyNet`'s derivative estimates in noisy regimes, and (2) our use of mathematically grounded derivative-based criteria rather than heuristics, which can fail in complex or borderline cases (see Appendix D). In full SR pipelines, robustness is further reinforced by our prior-based architecture: when separability detection becomes unreliable, the symbolic search defaults to standard RL behaviour rather than collapsing, whereas methods such as `AIF` and uDSR exhibit severe degradation under noise (with performance dropping by more than an order of magnitude at $10\%$ noise) (La Cava et al., 2021; Landajuela et al., 2022).

**Limitations and Future Directions** It should be emphasized that our method is particularly well-suited for physical science applications valuing interpretable exact solutions over numerical approximation, unlike SR approaches that prioritize accuracy at the expense of interpretability through long and intricate expressions[5]. In empirical sciences, considering model complexity alongside fit is especially valuable, and approaches based on BIC (Bayesian Information Criterion) or MDL (Minimum Description Length) have been developed to unify these criteria (Bartlett et al., 2023; Bastiani et al., 2024). In the context of structural discovery, this could motivate adaptive thresholds for separability detection to control the effective model complexity. However, unlike in standard SR, in structure discovery the quality of fit does not necessarily improve with increasing complexity.

**Conclusion** We introduced a SR framework that automatically discovers and exploits the underlying graph structure of physical data by identifying additive and multiplicative separabilities, including those nested within nonlinear transformations. This is enabled by a novel neural architecture and training scheme that provides a compact parametrization; the resulting strong regularization yields accurate derivatives, which are essential for detecting separabilities. This represents a significant advance over the previous structure discovery approach (`AIF`) which lacks these capabilities and consequently achieves much lower performances, particularly under noisy conditions.

The ability to uncover such structure inherently enhances interpretability and, importantly, the pre-processing step is general and can be integrated into any existing SR approach[6]. By incorporating this inductive bias as a prior within a reinforcement learning-based SR process, our method achieves state-of-the-art performance (72% exact recovery) on the Feynman benchmark from `SRBench`, marking a significant advance in physics-capable SR.

Notably, our method outperforms the seminal structure discovery approach `AIF`, which achieves 55% exact recovery, as well as the RL-hybrid `uDSR-A`, which builds on `AIF` and attains similar performance. It also surpasses the ensemble method `uDSR`, which integrates RL-based SR, `AIF`, large-scale pre-training, genetic programming, and other techniques, achieving 3% higher recovery than this combined approach. These results highlight how a robust mathematically grounded structure discovery module can dramatically facilitate SR.

## 5 REPRODUCIBILITY STATEMENT

An anonymized version of the code is available at [⊟ this link][7]. Comprehensive instructions for reproducing our experiments are provided in Appendix G. [○ On Github upon paper acceptance.]

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

## A    FORMULAS FOR PROPAGATING DIFFERENTIAL QUANTITIES IN COMPOSITE NESTYNET

In this appendix, we provide detailed formulas describing how differential information is propagated through a composite NestyNet model $f$, composed of neural sub-layers $\{f_i\}_{i=1}^N$ arranged in a tree structure, with each sub-layer operating on distinct subsets of the input. The overall target function can be expressed as

$$f = \Phi \circ (f_1, \dots, f_N),$$

where the composition function $\Phi$ hierarchically combines submodels via elementary operations (e.g., $+$, $\times$) or nonlinear transformations. An example of such a tree structure is shown in Panel a of Figure 1.

This structure differs fundamentally from classical neural networks in that we propagate full Jacobian information, i.e. derivatives not only with respect to model parameters but also with respect to each input data point $\{(\mathbf{x}, y)^{\langle k \rangle}\}_{k < n_{\text{samples}}}$. Here, $\mathbf{x} \in \mathbb{R}^{n_x}$ and $y \in \mathbb{R}$ (i.e., $n_y = 1$). This level of differential tracking is essential: it enables training the composite model using a Levenberg–Marquardt (LM) optimization procedure, which in turn allows us to achieve effective learning with relatively few parameters, a regularization which enables us to have very accurate derivative estimates.

The appendix is organized as follows. In Section A.1, we first describe a single sub-layer and provide its derivatives with respect to both data and model parameters. In Section A.2, we then enumerate all differential quantities required for evaluation and LM-based training, and give explicit formulas for their propagation throughout the composite tree.

### A.1    SINGLE LAYER

A single NestyNet layer is defined as:

$$y = a \, \log\big(1 + \exp(K\mathbf{x} + b)\big), \tag{7}$$

where $K \in \mathbb{R}^{h \times n_x}$ is a weight matrix, $b \in \mathbb{R}^h$ a bias vector, and $a \in \mathbb{R}^{n_y \times h}$ a scaling matrix. The set of trainable parameters is $\theta = \{K, b, a\} = \{\theta_i\}_{i < n_{\text{params}}}$.

This parametrization admits closed-form expressions for both the Jacobian and Hessian w.r.t. the inputs. Denoting by $\sigma(\cdot)$ the logistic sigmoid and by $\sigma'(\cdot) = \sigma(\cdot)\,(1 - \sigma(\cdot))$ its derivative:

$$\nabla_{\mathbf{x}} y = K^\top \big(a\,\sigma(K\mathbf{x} + b)\big), \tag{8}$$

$$\frac{\partial^2 y}{\partial x_i \, \partial x_j} = K^\top \big(a\,\sigma'(K\mathbf{x} + b)\big) K. \tag{9}$$

In addition, derivatives w.r.t. the model parameters $\{K, b, a\}$ can also trivially be written in closed form:

For the scaling matrix $a$:

$$\frac{\partial y}{\partial a} = \log\big(1 + \exp(K\mathbf{x} + b)\big)^\top, \tag{10}$$

with shape $\mathbb{R}^{n_y \times h}$, each entry $(i, j)$ corresponds to the sensitivity of output $y_i$ to the scaling parameter $a_{i,j}$.

For the bias vector $b$:

$$\frac{\partial y}{\partial b} = a \cdot \sigma(K\mathbf{x} + b), \tag{11}$$

with shape $\mathbb{R}^{n_y \times h}$, describing how variations in each bias term $b_j$ influence each output $y_i$, modulated by the scaling matrix $a$.

Finally, the derivative w.r.t. the weight matrix $K$ is

$$\frac{\partial y}{\partial K} = (a \cdot \sigma(K\mathbf{x} + b))\,\mathbf{x}^\top, \tag{12}$$

with shape $\mathbb{R}^{n_y \times h \times n_x}$, each slice along the last dimension corresponding to the contribution of an input variable $x_i$ to the sensitivity of $y$ w.r.t. the weight entries in the $i$-th column of $K$.

## A.2 Propagating differential quantities within a composite tree

Let us consider an arbitrary node within the tree : $f_v \in \{f_i\}_{i=1}^N$.

(i) $\underline{\text{Binary case:}}$ If $f$ has two children, $f_{v_1}$ and $f_{v_2}$, combined via a binary operation $f_v = g(f_{v_1}, f_{v_2})$, $g$ can be either addition ($f_v = f_{v_1} + f_{v_2}$) or multiplication ($f_v = f_{v_1}.f_{v_2}$).

(ii) $\underline{\text{Unary case:}}$ If $f_v$ has a single child, $f_{v_1}$, it is of the form $f_v = g(f_{v_1})$, where $g$ is a nonlinear transformation.

Here we focus on cases where the composition function is : $g \in \{+, \times, \square^{-1}, \square^2, \sqrt{\square}, \exp(\square)\}$. For each of these operations, we derive closed-form propagation rules for the differential quantities required in our framework:

- A.2.1 *Jacobians and Hessians*
    - Jacobians and Hessians w.r.t. inputs (for separability detection)
    - Jacobians w.r.t. model parameters (for LM optimization)

    I.e. Formulas for computing $J_{f_v}(\mathbf{x})$ from $J_{f_{v_1}}(\mathbf{x})$ (and $J_{f_{v_2}}(\mathbf{x})$ if applicable), and $H_{f_v}(\mathbf{x})$ from $H_{f_{v_1}}(\mathbf{x})$ (and $H_{f_{v_2}}(\mathbf{x})$).
- A.2.2 *Jacobian-vector products (JVP)*

    I.e., $J_{f_v}(\mathbf{x}) \cdot v$ from $J_{f_{v_1}}(\mathbf{x}) \cdot v$ (and $J_{f_{v_2}}(\mathbf{x}) \cdot v$).
- A.2.3 *Vector-Jacobian products (VJP, adjoints)*

    I.e., propagate $\bar{f}_v$ down to compute $\bar{f_{v_1}}$ (and $\bar{f_{v_2}}$).
- A.2.4 *Diagonal of the Gauss–Newton matrix*

    I.e. The diagonal of $J_{f_v}^\top J_{f_v}$ w.r.t. model parameters from $\mathrm{diag}(J_{f_{v_1}}^\top J_{f_{v_1}})$ (and $\mathrm{diag}(J_{f_{v_2}}^\top J_{f_{v_2}})$).

For clarity, we denote by $\theta = \{\theta_i\}_{i < n_{\text{params}}}$ the set of all parameters across the entire composite tree, i.e., the union of the parameters from all layer instances described in Section A.1. The total number of parameters in the tree is $n_{\text{params}}$.

### A.2.1 Jacobians and Hessians

Here we provide formulas for propagating Jacobians and Hessians throughout the composite tree. Depending on whether we want derivatives with respect to each input variable ($i < n_x$) or each model parameter ($i < n_{\text{params}}$), the dimensions of these quantities vary. For simplicity, we denote the second dimension (and third dimension for the Hessian) by $n_s$ to represent either $n_x$ or $n_{\text{params}}$ as appropriate. Then:

$$J_{f_v}(\mathbf{x}) \in \mathbb{R}^{n_{\text{samples}} \times n_s} \tag{13}$$

$$H_{f_v}(\mathbf{x}) \in \mathbb{R}^{n_{\text{samples}} \times n_s \times n_s} \tag{14}$$

(i) *Binary case:*

- Additive case: $f_v = f_{v_1} + f_{v_2}$

$$J_{f_v} = J_{f_{v_1}} + J_{f_{v_2}} \tag{15}$$

$$H_{f_v} = H_{f_{v_1}} + H_{f_{v_2}} \tag{16}$$

- Multiplicative case: $f_v = f_{v_1} \cdot f_{v_2}$

$$J_{f_v} = J_{f_{v_1}} \otimes f_{v_2} + J_{f_{v_2}} \otimes f_{v_1} \tag{17}$$

$$H_{f_v} = H_{f_{v_1}} \otimes f_{v_2} + H_{f_{v_2}} \otimes f_{v_1} + (J_{f_{v_1}})^\top \cdot J_{f_{v_2}} + (J_{f_{v_2}})^\top \cdot J_{f_{v_1}} \tag{18}$$

(ii) *Unary case:*

Here $g$ is univariate: $f_v = g(f_{v_1})$.

$$J_{f_v} = J_{f_{v_1}} \, g'(f_{v_1}) \,, \tag{19}$$

$$H_{f_v} = H_{f_{v_1}} \, g'(f_{v_1}) + J_{f_{v_1}}^\top J_{f_{v_1}} \, g''(f_{v_1}) \,. \tag{20}$$

where $g'$ denotes the derivative of $g$ with respect to its input.

### A.2.2 Jacobian-Vector Products (JVP)

The Jacobian-vector product (JVP) encodes the directional derivative of $f_v$ along a vector $\nu \in \mathbb{R}^{n_x}$. It is defined as:

$$D_\nu f_v = J_{f_v} \cdot \nu \tag{21}$$

where $J_{f_v} \in \mathbb{R}^{\text{samples} \times n_x}$ and $D_\nu f_v \in \mathbb{R}^{n_{\text{samples}}}$.

(i) *Binary case:*

- Additive case: $f_v = f_{v_1} + f_{v_2}$

$$D_\nu f_v = D_\nu f_{v_1} + D_\nu f_{v_2} \tag{22}$$

- Multiplicative case: $f_v = f_{v_1} \cdot f_{v_2}$

$$D_\nu f_v = D_\nu f_{v_1} \cdot f_{v_2} + D_\nu f_{v_2} \cdot f_{v_1} \tag{23}$$

(ii) *Unary case:*

Here $g$ is a univariate function: $f_v = g(f_{v_1})$. The JVP follows the chain rule:

$$D_\nu f_v = g'\left(f_{v_1}\right) \cdot D_\nu f_{v_1} \tag{24}$$

### A.2.3 Vector-Jacobian Products (VJP, adjoints)

The adjoint of a node $f_v$ encodes the sensitivity of the root node $f$ with respect to $f_v$, weighted by a vector $\omega \in \mathbb{R}^{n_{\text{samples}}}$:

$$\bar{f}_v = \frac{\partial f}{\partial f_v}\,\omega \tag{25}$$

where $f$ is the output at the root of the tree, $f_v$ is the value at the current node, and $\omega$ is a vector representing the incoming sensitivity.

Unlike standard Jacobian propagation, adjoints propagate sensitivities *top-down* through the tree. That is, given the adjoint of a parent node $f_v$, we want to compute the adjoint of a child node $f_{v_1}$ (which may have a sibling $f_{v_2}$ in the binary case).

Formally, the adjoint of a child node can be expressed as a function of its parent:

$$\bar{f_{v_1}} = \frac{\partial f}{\partial f_{v_1}}\,\omega = \frac{\partial f}{\partial f_v}\frac{\partial f_v}{\partial f_{v_1}}\,\omega = \frac{\partial f_v}{\partial f_{v_1}}\,\bar{f}_v. \tag{26}$$

(i) *Binary parent:*

- Additive parent: $f_v = f_{v_1} + f_{v_2}$

$$\bar{f_{v_1}} = \bar{f}_v, \quad \bar{f_{v_2}} = \bar{f}_v \tag{27}$$

- Multiplicative parent: $f_v = f_{v_1} \cdot f_{v_2}$

$$\bar{f_{v_1}} = \bar{f}_v \cdot f_{v_2}, \quad \bar{f_{v_2}} = \bar{f}_v \cdot f_{v_1} \tag{28}$$

(ii) *Unary parent:* Here $g$ is a univariate function: $f_v = g(f_{v_1})$. The adjoint propagates via the chain rule:

$$\bar{f_{v_1}} = \bar{f}_v \cdot g'(f_{v_1}). \tag{29}$$

### A.2.4 Diagonal of the Gauss–Newton Matrix

The diagonal of the Gauss–Newton matrix is defined as

$$\text{diag}\left(J_{f_v}^\top J_{f_v}\right), \tag{30}$$

where $J_{f_v}$ is the Jacobian of $f_v$ with respect to all model parameters $\theta$ (across all layers). This diagonal encodes the squared sensitivities of the output with respect to each parameter. Importantly, for binary compositions, cross-terms of the form $\frac{\partial f_{v_1}}{\partial \theta_i} \cdot \frac{\partial f_{v_2}}{\partial \theta_i}$ vanish because each parameter $\theta_i$ belongs to exactly one child, either $f_{v_1}$ or $f_{v_2}$, but not both. This property allows efficient propagation of the diagonal without computing the full $(J^\top J) \in \mathbb{R}^{n_{\text{samples}} \times n_{\text{params}}}$.

(i) *Binary parent:*

- Additive parent: $f_v = f_{v_1} + f_{v_2}$

$$\text{diag}\left(J_{f_v}^\top J_{f_v}\right) = \text{diag}\left(J_{f_{v_1}}^\top J_{f_{v_1}}\right) + \text{diag}\left(J_{f_{v_2}}^\top J_{f_{v_2}}\right) \tag{31}$$

- Multiplicative parent: $f_v = f_{v_1} \cdot f_{v_2}$

$$\text{diag}\left(J_{f_v}^\top J_{f_v}\right) = f_{v_2}^2 \cdot \text{diag}\left(J_{f_{v_1}}^\top J_{f_{v_1}}\right) + f_{v_1}^2 \cdot \text{diag}\left(J_{f_{v_2}}^\top J_{f_{v_2}}\right) \tag{32}$$

(ii) *Unary parent:* For a univariate function $g$, $f_v = g(f_{v_1})$, the diagonal propagates as

$$\text{diag}\left(J_{f_v}^\top J_{f_v}\right) = \left(g'(f_{v_1})\right)^2 \cdot \text{diag}\left(J_{f_{v_1}}^\top J_{f_{v_1}}\right). \tag{33}$$

## B  COMPOSITE NESTYNET TRAINING AND HYPER-PARAMETERS

As detailed in Section 2, our approach involves fitting a composite tree of `NestyNet` layers. Here we summarize the key hyper-parameters and optimizer settings used.

For the Feynman benchmark, we sample 10,000 points and split them equally into training and test sets, with a batch size of 5,000.

Neural network parameters are stored in `float64` and initialized randomly with a scale of 0.1. Each node in the tree uses a hidden layer of size $h = 16$ with a single layer per node. If higher accuracy is required, we increase the hidden size to $h = 48$ and use two layers per node. The model is trained for 5,000 iterations using the Levenberg–Marquardt (LM) optimizer.

We employ a damped LM scheme with an initial damping $\lambda_0 = 10^2$, a decrease factor of 3, an increase factor of 5, and bounds $\lambda_{\min} = 10^{-10}$ and $\lambda_{\max} = 10^{12}$.

## C  DETAILS ABOUT REINFORCEMENT LEARNING-BASED SYMBOLIC REGRESSION

We summarize here the reinforcement learning (RL) setup used for the symbolic regression (SR) stage of our method – corresponding to panel b of Figure 1. For completeness, we describe the priors, architecture, reward definition, and optimization strategy.

**Priors.** In addition to the structural prior that is the object of this study and the associated per-subexpression length prior, we employ several additional constraints to guide expression generation:

- Maximum expression length of 50 tokens.
- Soft length prior: a Gaussian with mean 12 and variance $\sigma^2 = 5$, encouraging concise expressions.
- Nested trigonometric operations are limited to two levels (e.g., forbidding $\cos(f \cdot t + \sin(x/x_0 + \tan(\square)))$, but allowing $\cos(f \cdot t + \sin(x/x_0))$).
- Exponential and logarithmic operators cannot be self-nested (e.g., forbidding $e^{e^\square}$).
- Inverse unary operations that cancel each other are forbidden (e.g., forbidding $e^{\log \square}$).
- Dimensional analysis prior: ensures that generated expressions are dimensionally consistent.

When conflicting priors arise, the corresponding candidate expression is discarded.

**Architecture.** We use an LSTM-based policy network that observes the context around the next token to be generated. At each generation step, the network receives:

- the parent and sibling tokens, along with their units,
- the previously sampled token,
- the required units for the next token, and

- the dangling number, i.e., the minimum number of remaining tokens required for a valid expression.

Based on this information, the LSTM outputs a categorical distribution over the token library and a hidden state that is passed to the next step. The distribution is then modified according to the priors described above. This process is repeated until a complete mathematical expression is sampled, as illustrated in Figure 4.

**Reward Function.** Once an expression is generated, free constants are optimized using `LBFGS` with automatic differentiation. The reward—representative of the fit quality—of a candidate expression $f$ is defined as

$$R(f) = \frac{1}{1 + \text{NRMSE}(f)}, \tag{34}$$

$$\text{NRMSE}(f) = \frac{1}{\sigma_y} \sqrt{\frac{1}{n_{\text{samples}}} \sum_{k=1}^{n_{\text{samples}}} \left(y^{\langle k \rangle} - f(\mathbf{x}^{\langle k \rangle})\right)^2}, \tag{35}$$

where $\sigma_y$ is the standard deviation of the target values. The reward is normalized to $[0, 1]$, with $R = 1$ indicating a perfect fit.

**Reinforcement Learning Optimization.** Policy gradients are approximated using the top $5\%$ of candidates, following the risk-seeking strategy of Petersen et al. (2021), inspired by Rajeswaran et al. (2017). Entropy regularization from Landajuela et al. (2021) is applied to encourage exploration, with weight $0.005$ and sequence-dependent decay $\gamma^t$ at step $t$, with $\gamma = 0.7$. The training batch size is set to 10,000, and parameters are updated using the Adam optimizer with a learning rate of $0.0025$.

## D    ABOUT MULTIPLICATIVE SEPARABILITY DETECTION IN `AIF`

The `AIF` method introduced in Udrescu & Tegmark (2020); Udrescu et al. (2020) aims to detect additive and multiplicative separability in data pairs $(\mathbf{x}, y)$ as a pre-processing step for symbolic regression (SR). Including by uncovering separable structures of the form

$$y = f(x_1, \ldots, x_{n_x}) = f_1(x_1, \ldots, x_j) \cdot f_2(x_1, \ldots, x_{n_x}),$$

the SR problem can be recursively reduced to simpler subproblems. Although our approach shares the overarching objective of leveraging separability to reduce SR complexity, the underlying methodology and criteria differ fundamentally, resulting in a substantial performance gap between the two methods.

**Neural emulator** In `AIF`, a multilayer perceptron (MLP) is first trained to approximate the target function, yielding an emulator $f_{\text{NN}}$ such that $y \approx f_{\text{NN}}(\mathbf{x})$. Separability tests are then carried out using this neural approximation. Our approach also uses a neural emulator, but replaces the MLP with `NestyNet`, which provides significantly more accurate derivative estimates (see Section 2.1). This is essential because our separability criteria explicitly rely on first- and second-order partial derivatives of $y$ with respect to the input variables (Section 2.2). These derivative-based criteria yield mathematically grounded conditions for multiplicative separability.

**`AIF` does not use derivatives for detecting multiplicative separability** This is the most important conceptual distinction. Although Udrescu et al. (2020) mention a derivative-based criterion (e.g. their Eq. 6), the *actual* `AIF` implementation does *not* use derivatives to assess multiplicative separability. Instead, it applies a heuristic based purely on neural emulator evaluations. This can be verified in their public implementation: see the function `check_separability_multiply` in [ aifeynman/S_separability.py][8] (Line 213).

**The `AIF` heuristic for multiplicative separability** When testing whether two groups of variables $\mathbf{x}_1$ and $\mathbf{x}_2$ are multiplicatively separable, `AIF` checks whether the emulator satisfies, approximately,

$$f_{\text{NN}}(\mathbf{x}_1, \mathbf{x}_2) \approx f_{\text{NN}}(\mathbf{x}_1, \text{med}(\mathbf{x}_2)) \, f_{\text{NN}}(\text{med}(\mathbf{x}_1), \mathbf{x}_2),$$

that is, whether freezing each variable group at its median value yields factors whose product approximates the original prediction.

---

[8]`https://github.com/SJ001/AI-Feynman/blob/master/aifeynman/S_`
`separability.py#L213`

| # | Test case | **Ours** | AIF |
|---|-----------|:--------:|:---:|
| 1 | $y = x_1 x_2$ | ✓ | ✓ |
| 2 | $y = e^{-x_1} \cdot x_2$ | ✓ | ✓ |
| 3 | $y = \cos(x_1) \cdot \arctan\left(\frac{1}{x_2}\right)$ | ✓ | ✓ |
| 4 | $y = \cos\left(\frac{1}{x_1}\right) \cdot \sin\left(\frac{1}{x_2}\right)$ | ✓ | |
| 5 | $y = \cos(x_1 x_2) \cdot \frac{1}{x_3}$ | ✓ | |
| 6 | $y = e^{-x_1 x_2} \tan(x_3)$ | ✓ | |
| 7 | $y = (x_1 + x_2) \cdot \cos(x_3 x_4)$ | ✓ | |
| 8 | $y = (x_1 + x_2) \cdot \frac{1}{x_3 + x_4}$ | ✓ | |
| 9 | $y = (x_1 + x_2) \cdot (x_3 + x_4)$ | ✓ | ✓ |
| 10 | $y = \frac{x_1}{x_2} \cdot \frac{x_3}{x_4}$ | ✓ | |
| 11 | $y = (x_1 + x_2)^2 \cdot (x_3 + x_4)^2$ | ✓ | |
| 12 | $y = \cos(x_1 + x_2) \cdot \log(x_3 + x_4)$ | ✓ | |

Table 4: **`AIF`'s limitations in multiplicative separability detection.** Comparison of multiplicative separability detection performances on synthetic data from 12 test equations. Our derivative-based criterion correctly identifies multiplicative separability in all cases, whereas `AIF` succeeds only in the simplest scenarios, predominantly those involving two variables or near-linear structures.

The score used in `AIF` is

$$\epsilon_{\text{mul,AIF}} = \underset{y > 0.2 \cdot \max(y)}{\text{med}} \left( 2 \left| y - \frac{f_{\text{NN}}(\mathbf{x}_1, \mathbf{x}_2)}{f_{\text{NN}}(\mathbf{x}_1, \text{med}(\mathbf{x}_2)) \, f_{\text{NN}}(\text{med}(\mathbf{x}_1), \mathbf{x}_2)} \right| \right).$$

Unlike our metric, this heuristic does not derive from the mathematical definition of multiplicative separability; it depends heavily on the emulator's behavior under variable freezing and therefore exhibits inconsistent performance.

**Performance of `AIF` on multiplicative separability** The limitations of this heuristic are evident when evaluated on test cases where `AIF` often fails to recover multiplicative structure except in trivially simple cases. This is notably reflected by its performance on the Feynman Benchmark (Fig. 5).

We further tested `AIF` on 12 synthetic multiplicatively separable equations (each with 1000 samples uniformly drawn in $[0.1, 1]$) and compared its results with our derivative-based criterion in Table D. For fairness, only additive and multiplicative separability tests were enabled in both methods. The results confirm that `AIF` succeeds only in trivial scenarios (e.g. linear or two-variable cases), while our method recovers the correct separability across all tested equations.

**`AIF` cannot return true negatives** Finally, it is important to note that `AIF` was developed along-side—and for—the Feynman Benchmark, where all target functions are known *a priori* to be separable. Consequently, `AIF` always returns a separability type: it computes scores for all separability hypotheses and selects the one with the smallest error, regardless of whether the data are truly separable. This design makes `AIF` unsuitable for real-world data where no separability may exist.

In principle, `AIF` cannot assert the *absence* of separability, a limitation that may lead to false positives. To remain conservative in our comparison, we consider `AIF` to have failed only when it assigns a better score to additive separability on data that are in fact multiplicatively separable. Even under this generous criterion, `AIF` performs poorly. Because `AIF` cannot detect true negatives by design, we refrain from conducting tests involving non-separable functions.

# E  SEPARABILITY BENCHMARK

**Separability Detection Benchmark** We introduce a benchmark specifically designed to evaluate separability detection methods. The goal is to determine, given a dataset $\{\mathbf{x}, y\}$, whether it can be decomposed according to distinct partitions of input variables. That is, instead of modeling $y = f(\mathbf{x})$, the target may be representable as $y = f_a(\mathbf{x}_1) + f_b(\mathbf{x}_2)$ or $y = f_a(\mathbf{x}_1) \cdot f_b(\mathbf{x}_2)$, where $\mathbf{x}_1$ and $\mathbf{x}_2$ are disjoint subsets of $\mathbf{x}$.

The benchmark comprises 24 challenges:

- Some exhibit additive separability, while others are multiplicatively separable.
- Certain separabilities are embedded within nonlinear transformations $g(\cdot)$.
- Some challenges contain no separability, ensuring that methods can correctly detect its absence.

For each challenge, a ground-truth equation is used to generate a synthetic dataset. The task is to recover, using only the dataset, whether a separability exists between subsets of input variables. We expect this benchmark to serve as a valuable tool for the community, providing a standardized way to evaluate separability detection methods. Table E summarizes each challenge and reports our method's recovery rates.

**Protocol** For each challenge, we generate 10,000 samples with input values drawn uniformly from the interval $[0.1, 2]$. Gaussian noise is added to the output $y$ as follows:

$$y_{\text{noisy}} = y + \epsilon, \quad \epsilon \sim \mathcal{N}\Big(0, \gamma \sqrt{\frac{1}{N} \sum_{i=1}^{N} y_i^2}\,\Big), \tag{36}$$

where $N$ is the number of samples and $\gamma$ specifies the fractional noise level. Each challenge is evaluated at four noise levels ($0\%$, $1\%$, $5\%$, and $10\%$) and repeated across four random seeds.

**Separability Threshold** As described in Sub-section 2.2, our separability detection relies on three quantities that become negligible when a true separation exists. To assess this negligibility, we defined threshold values $\epsilon_{\text{add}}$, $\epsilon_{\text{mul}}$, and $\epsilon_{\beta_{\text{mad}}}$, below which the criteria are considered satisfied and provided their values for noiseless scenarios in Table 2.

In the presence of noise, these thresholds are relaxed proportionally to the observed noise level, as it is normal for quantities to appear only approximately negligible. We estimate the effective noise by evaluating the fit quality of our neural emulator $f_{\text{NN}}$, which is also used to compute derivatives, using the root mean squared error (RMSE) on the dataset $y$. The adaptive thresholds are then given by:

$$\epsilon = \min\big(\alpha \cdot \text{RMSE} + \epsilon_{\text{base}}, \ \epsilon_{\text{max}}\big) \tag{37}$$

Here, $\epsilon_{\text{base}}$ is the base threshold from Table 2, $\epsilon_{\text{max}}$ is the maximum allowed relaxation, and $\alpha$ is a tunable coefficient controlling sensitivity to noise. The RMSE is computed as:

$$\text{RMSE} = \sqrt{\frac{1}{N} \sum_{i=1}^{N} \big(f_{\text{NN}}(\mathbf{x}_i) - y_i\big)^2}. \tag{38}$$

We optimize the adaptive threshold parameters using a Newton-based optimization scheme, resulting in $\alpha = 4$ and $\epsilon_{\text{max}} = 0.1$, which we found to balance sensitivity and robustness effectively across our benchmark.

**Computational cost.** Each separability test, including training the neural emulator and analyzing its derivatives, requires approximately 5 minutes per test when run on 16 cores of an AMD EPYC 7532 CPU.

| Separability | | # | Data | Ground Truth | Noise | | | |
|---|---|---|---|---|---|---|---|---|
| | | | | | 0 % | 1 % | 5 % | 10 % |
| Direct separability | $f_a + f_b$ | 1 | $x_1 + x_2$ | $f_a(x_1) + f_b(x_2)$ | 100 | 100 | 100 | 100 |
| | | 2 | $x_1 x_2 + x_3 x_4$ | $f_a(x_1, x_2) + f_b(x_3, x_4)$ | 100 | 100 | 100 | 100 |
| | | 3 | $(x_1 - x_2)^2 + (x_3 - x_4)^2$ | $f_a(x_1, x_2) + f_b(x_3, x_4)$ | 100 | 100 | 100 | 100 |
| | | 4 | $x_2 e^{-x_1} + \sin(x_3 x_4)$ | $f_a(x_1, x_2) + f_b(x_3, x_4)$ | 100 | 25 | 0 | 0 |
| | $f_a \cdot f_b$ | 5 | $x_1 x_2$ | $f_a(x_1) f_b(x_2)$ | 100 | 100 | 100 | 100 |
| | | 6 | $\frac{x_1}{x_2}$ | $f_a(x_1) f_b(x_2)$ | 100 | 75 | 100 | 75 |
| | | 7 | $\frac{x_1 + x_2}{x_3 + x_4}$ | $f_a(x_1, x_2) f_b(x_3, x_4)$ | 100 | 100 | 0 | 0 |
| | | 8 | $(x_1 + x_2 + x_3)(x_4 + x_5 + x_6)$ | $f_a(x_1, x_2, x_3) f_b(x_4, x_5, x_6)$ | 100 | 100 | 100 | 50 |
| | | 9 | $x_2 e^{-x_1}$ | $f_a(x_1) f_b(x_2)$ | 100 | 100 | 100 | 100 |
| | | 10 | $x_1 x_2^2$ | $f_a(x_1) f_b(x_2)$ | 100 | 100 | 100 | 75 |
| | | 11 | $e^{-x_1} \cos(x_2 + x_3)$ | $f_a(x_1) f_b(x_2, x_3)$ | 100 | 100 | 100 | 100 |
| | | 12 | $x_1 \log(x_2 + x_3)$ | $f_a(x_1) f_b(x_2, x_3)$ | 100 | 100 | 100 | 100 |
| | | 13 | $\frac{e^{-x_1}}{x_2 + x_3}$ | $f_a(x_1) f_b(x_2, x_3)$ | 100 | 100 | 50 | 0 |
| | | 14 | $\log(x_3 + x_4) \cos(x_1 + x_2)$ | $f_a(x_1, x_2) f_b(x_3, x_4)$ | 100 | 100 | 100 | 100 |
| | | 15 | $(x_1 + x_2)^2 (x_3 + x_4)^2$ | $f_a(x_1, x_2) f_b(x_3, x_4)$ | 100 | 100 | 100 | 100 |
| Nested separability | $g(f_a + f_b)$ | 16 | $\cos(x_1 + x_2)$ | $\cos(f_a(x_1) + f_b(x_2))$ | 100 | 75 | 100 | 25 |
| | | 17 | $(x_1 + x_2)^2$ | $(f_a(x_1) + f_b(x_2))^2$ | 100 | 100 | 100 | 75 |
| | | 18 | $\sqrt{x_1 + x_2}$ | $\sqrt{f_a(x_1) + f_b(x_2)}$ | 100 | 75 | 25 | 0 |
| | | 19 | $\sqrt{(x_1 + x_2)^2 + (x_3 + x_4)^2}$ | $\sqrt{f_a(x_1, x_2) + f_b(x_3, x_4)}$ | 100 | 100 | 100 | 100 |
| | | 20 | $\sqrt{x_1^2 + x_2^2}$ | $\sqrt{f_a(x_1) + f_b(x_2)}$ | 100 | 100 | 75 | 100 |
| | $g(f_a \cdot f_b)$ | 21 | $\cos(x_1 x_2)$ | $\cos(f_a(x_1) f_b(x_2))$ | 100 | 100 | 50 | 75 |
| No separability | | 22 | $x_1 \log(x_1 + x_2)$ | $f(x_1, x_2)$ | 100 | 100 | 100 | 100 |
| | | 23 | $x_3 + \frac{\cos(x_1 + x_2)}{x_2 + x_3}$ | $f(x_1, x_2, x_3)$ | 100 | 100 | 100 | 100 |
| | | 24 | $\frac{\sqrt{\frac{x_1^2}{x_2^2} + 1}}{\frac{x_1 \cos(x_3)}{x_2} + 1}$ | $f(x_1, x_2, x_3)$ | 100 | 100 | 100 | 100 |
| | | | | | **100.0** | **93.8** | **78.3** | **76.5** |

Table 5: **Separability benchmark.** The benchmark comprises 24 challenges, including additive and multiplicative separabilities, potentially nested within nonlinear functions, as well as cases with no separability to assess false-positive resilience. The goal of each challenge is to recover the correct separability (ground-truth column) from data samples generated from the equation in the data column at various noise levels. Recovery performance of our method is reported in % under various noise levels.

## F    FEYNMAN BENCHMARK

We evaluated our method on the Feynman benchmark, which comprises 120 equations (mainly sourced from the *Feynman Lectures on Physics* (Feynman et al., 1971)) to be exactly recovered from their associated data. This benchmark was initially introduced by Udrescu & Tegmark (2020) and subsequently standardized and formalized through the widely used SRBench framework (La Cava et al., 2021).

**Protocol.** We strictly follow the protocol outlined in (La Cava et al., 2021), excluding challenges I.26.2, I.30.5, II.11.17, and test_10, leaving a total of 116 challenges for evaluation. Comparisons with baseline methods were performed only on these selected challenges. Exact symbolic recovery was assessed using SymPy's (Meurer et al., 2017) equivalence checking. To ensure consistency, we used the same hyper-parameters across all challenges. We performed a recursive structure analysis to identify separable components, including ones potentially nested in nonlinearities $g_{\mathrm{NL}} \in \{\Box^{-1}, \Box^2, \sqrt{\Box}, \exp, \log, \cos\}$. The detected structure was then used as a prior for RL-based symbolic regression, permitting the operators $\{+, -, \times, /, 1/\Box, \sqrt{\Box}, \Box^2, -\Box, \exp, \log, \cos, \sin\}$ and two dimensionless free constants plus a constant equal to one, $\{\theta_1, \theta_2, 1\}$ for composing trial expressions. We strictly followed the established benchmarking protocol, using datasets of 10,000 points and limiting the maximum number of expression evaluations per challenge to 1 million.

**Computational Cost** The pre-processing structure analysis requires $\sim 5$ minutes to train a composite NestNet and assess separability [9] on an Nvidia GV100. Depending on the problem dimensionality and the number of separabilities identified, the total analysis time varies substantially from $\sim 1$ to $\sim 24$ hours., averaging about 4 hours per Feynman benchmark case.

For RL-based SR, the dominant cost—as is common in trial-and-error SR—lies in free constant fitting for evaluating candidate expressions. This step is more efficient on CPUs and thus primarily CPU-bound. On the Feynman benchmark, a typical case takes about 4 hours when utilizing all cores of an Intel Xeon W-2155 CPU.

## G    CODE AND REPRODUCIBILITY DETAILS

**Repository** Our code is fully open source, and a link to the GitHub repository will be provided upon paper acceptance. To ensure long-term reproducibility, we will also release a frozen version of the code corresponding to this paper. In the meantime, an anonymized version of the code is available at [🖹 this link][10]. [○ Update upon paper acceptance.]

**Reproducibility.** For the sake of result reproducibility, we offer a simple method to replicate the outcomes presented in Figure 5 that reflects the two stage process described in Figure 1.

(a) *Structure analysis.* Run

```
python feynman_struct_run.py --equation i
```

where i $\in \{0, 1, \ldots, 119\}$ corresponds to the equation index in the Feynman benchmark. This produces the file structure_analysis.csv containing the detected separabilities.

(b) *Symbolic regression.* In the same folder, run

```
python feynman_run.py --equation i -c feynman_config_r12
```

which performs RL-based symbolic regression using the structure discovered in step (a).

---

[9]For an individual separability test, we evaluate derivative-based metrics for all variable pairs, giving $O(n_x{}^2)$ complexity for $n_x$ input variables. For $n_x \leq 10$ this means $\leq 100$ Hessian checks, all performed in inference mode which is computationally inexpensive compared to other steps of the method

[10]drive.google.com/drive/folders/1ABm2kGdbVQG54dhAZZR1lgdA_zbgrUve?usp=drive_link

Finally, results can be aggregated and analyzed via:

```
python feynman_results_analysis.py
```

