# OpenReview forum: "Recursive Structure Discovery as an Inductive Bias for Symbolic Regression"
_ICLR.cc/2026/Conference — Submitted to ICLR 2026_

### Official Review · Reviewer_9hzL · 2025-10-30

**Soundness:** 3
**Presentation:** 2
**Contribution:** 1
**Rating:** 2
**Confidence:** 4

**Summary:**

This paper proposes a two-stage framework: it first leverages derivative information to uncover the variable structure within expressions, then performs variable separation to decompose the overall problem into subproblems, and finally applies a reinforcement learning–based symbolic regression algorithm. The method achieves a high recovery rate on SRBench.

**Strengths:**

The ideas in the paper are sound, the language has no obvious issues, and the overall structure is complete.

**Weaknesses:**

The literature review is insufficient. Derivative-based variable separation was first introduced into symbolic regression by AI Feynman, and AI Feynman 2.0 already handles variable separation for most separable cases, including multiplicative separability.

Udrescu, S. M., Tan, A., Feng, J., Neto, O., Wu, T., & Tegmark, M. (2020). AI Feynman 2.0: Pareto-optimal symbolic regression exploiting graph modularity. Advances in Neural Information Processing Systems, 33, 4860-4871.

The paper lacks novelty. The proposed method merely stitches together AI Feynman and DSR, and this combination has already been explored by other researchers, such as the UDSR mentioned in the paper.

The experimental baselines omit recent strong methods. The experiments lack reproducible and convincing empirical support; the reported performance of some baselines differs markedly from results in other papers and from those obtained using the corresponding open-source implementations.

**Questions:**

None.

---

> ### Author Response · Authors · 2025-11-23
> **AI Feynman does **not** use derivative-based multiplicative separability**
>
> Thank you for the pertinent review. Before starting this project, we actually shared the reviewer’s intuition: we also believed that **AI Feynman 2.0 (AIF 2.0)** implemented *derivative-based* variable separability, including multiplicative separability. However, this is **not the case**, and we are grateful for the opportunity to clarify this important point.
>
> **1. AIF 2.0 does *not* use derivative-based multiplicative separability**
>
> Although the authors of AIF 2.0 *suggest* using derivatives for multiplicative separability (see Eq. (6) in [AIF 2.0](https://arxiv.org/abs/2006.10782)), their **actual implementation** relies on a very fragile heuristic. Specifically, the function `check_separability_multiply` in their public code  [github.com/SJ001/AI-Feynman/aifeynman/S_separability.py](https://github.com/SJ001/AI-Feynman/blob/master/aifeynman/S_separability.py#L213) (Line 213) checks whether the neural network emulator  $f_{\text{NN}}$ satisfies:
> $$
> f\_{\text{NN}}(\mathbf{x}\_1,\mathbf{x}\_2)\approx
> f\_{\text{NN}}(\mathbf{x}\_1,\text{med}(\mathbf{x}\_2)) \\cdot
> f\_{\text{NN}}(\text{med}(\mathbf{x}\_1),\mathbf{x}\_2).
> $$
>
> This procedure is **not** a mathematically grounded criterion for multiplicative separability.
> This limitation explains in part the poor performance of AIF 2.0 on SRBench (55% recovery vs. 72% for ours).
>
> We now also demonstrate this empirically via a controlled comparison on data from 12 synthetic test equations (Table 3), showing that AIF 2.0 is able to detect multiplicative separability only in trivial cases (e.g., linear or two-variable scenarios), whereas our method succeeds on all 12. This robustness stems from our criterion being derived from the mathematical derivative properties of the multiplicative operator (as detailed in Subsection 2.2 of our paper).
>
> We suspect that AIF 2.0 could not implement derivative-based criteria because it relies on MLPs, which approximate functions well but are poor estimators of derivatives with respect to input variables.
>
> We now state these points explicitly in the literature review and in a dedicated analysis in *Appendix D: About multiplicative separability detection in AIF*.
>
> **2. Novelty**
>
> Our method is *not* a simple “stitching” of [DSR](https://arxiv.org/abs/1912.04871) and [AIF 2.0](https://arxiv.org/abs/2006.10782), nor is it comparable to the hybridization done in [uDSR](https://openreview.net/forum?id=2FNnBhwJsHK).
> Our structure-analysis pipeline introduces multiple major innovations not present in AIF 2.0, DSR, or uDSR:
>
> **a)** A mathematically grounded, **derivative-based multiplicative separability criterion**.
> **b)** Detection of **nested separability inside nonlinear functions**.
> **c)** Use of structural analysis as a **soft inductive bias** for RL-based SR, rather than a rigid constraint.
> **d)** Integration of a **dimensional-analysis prior**, uniquely combined with (a)–(c).
>
> This combination produces capabilities not available in prior work and explains our performance improvement over DSR, AIF 2.0, and uDSR. The **DSR/AIF only ablation of uDSR achieves 50% on the Feynman benchmark vs. 72% for our method** . To clarify this, we expanded the literature review and added a **feature comparison table (Table 1).**
>
> **3. Baseline performance**
>
> Fourteen of our baseline performance values come directly from **[SRBench](https://arxiv.org/abs/2107.14351)**, whose numbers are widely used in the community (e.g., in [ParFam](https://arxiv.org/abs/2310.05537), [E2E SR](https://arxiv.org/abs/2204.10532), [DGSR](https://arxiv.org/abs/2401.00282), [CADSR](https://arxiv.org/abs/2406.06751), [SR-GPT](https://arxiv.org/abs/2401.14424), [SymQ](https://arxiv.org/abs/2402.05306v1), [FormulaGPT](https://arxiv.org/abs/2404.06330), [SNIP](https://arxiv.org/abs/2310.02227), …).
> Four additional numbers come from [uDSR](https://openreview.net/forum?id=2FNnBhwJsHK) and [LaSR](https://arxiv.org/abs/2409.09359).
>
> A likely source of confusion is that **we evaluate on all 120 Feynman problems**, whereas some other works report results only on the easiest 100. We now clarify this clearly in the caption of Figure 5. If the reviewer believes additional baselines should be included, we would be happy to add them.
>
> **4. Reproducibility**
>
> We provide:
>
> - complete methodological details,
> - every hyperparameter in Table 2 and Appendices B–C,
> - fully open-source code (link in Section 5),
> - and a two-line command to reproduce any of the 120 experiments (Appendix E).
>
> We are confident that the work is fully reproducible. We note that ICLR guidelines recommend placing detailed reproducibility information in an appendix, which is why Section 5 primarily directs the reader to Appendix E. We are happy to provide any additional details that the reviewer may consider helpful.
>
> **5. Implemented changes**
>
> All changes in the paper are highlighted in **red** for clarity.
> The new experiments we ran for the purpose of this review are in App. D.

---

> ### Author Response · Authors · 2025-11-28
> **Noise Robustness & AIF**
>
> As requested by reviewers, we conducted additional experiments (see updated Sec. 3 and Discussion) to evaluate the noise resilience of our method. The results demonstrate that our approach is **highly robust towards noise**: it retains approximately 75% of its detection capability at 10% Gaussian noise. Perhaps even more importantly, it exhibits extreme **conservatism with respect to false positives**: while high noise can occasionally obscure true separabilities, our method never reports a separability where none exists.
>
> This performance stands in stark contrast to **AIF 2.0, which does not employ derivative-based criteria**. In noisy conditions, AIF’s performance collapses—from 55% exact recovery at 0% noise to less than 1% at 10% noise on SRBench. Similarly, uDSR, which incorporates AIF, drops from 69% to 11% due to noise, performing even worse than its simpler DSR component due to the reliance on AIF.
>
> We attribute the robustness of **our approach to the mathematically grounded, derivative-based separability criteria**, which provide reliable detection even in challenging noisy scenarios. These results highlight the practical advantage of our structure-discovery pipeline over previous methods.

---

### Official Review · Reviewer_zTXN · 2025-10-31

**Soundness:** 2
**Presentation:** 2
**Contribution:** 3
**Rating:** 4
**Confidence:** 3

**Summary:**

This paper proposes a recursive structure discovery framework that improves symbolic regression (SR) by uncovering how variables in scientific data combine through simple, interpretable structures. Instead of directly fitting equations, the method first trains a compact NestyNet with accurate estimated derivatives. These derivatives are used to detect additive, multiplicative, and nested nonlinear separabilities, building a hierarchical tree of sub-functions. This structure acts as a probabilistic prior guiding a deep reinforcement learning-based SR model to generate mathematically consistent expressions. The proposed method is demonstrate to outperform many existing SR methods.

**Strengths:**

1. This paper proposes a novel approach to first use powerful neural networks to fit the data and then detect key structures and inform symbolic regression.
2. This paper proposes the key issue of compactness of the learned expression and the benefit of doing so, which may inspire later work.
3. The empirical performance is convincing.

**Weaknesses:**

1. The paper is a bit hard to read from time to time, possibly because there are many components in the method (NestyNet, finding structures, and incorporating it into symbolic regression).
2. How the tree is built is not clear -- it might be helpful to elaborate on this. I understand that $f_i$ is a NestyNet, but it is unclear how the mapping $\Phi$ is determined. The paragraph "Composite model" on page 5 is difficult to read.
3. It's unclear how the threshold values are chosen in Table 1 (and there is a typo `treshold' above it).
4. It is unclear why the method is paired with the RNN method.

**Questions:**

1. Can you explain how the composite model on page 5 is trained?
2. Why would you use a tree structure to combine the NestyNet layers? It is also common to use tree to represent expressions in symbolic regression, but it's unclear to me why using a tree makes sense here.
3. How did you choose the threshold values in Table 1?
4. How does the method compare with uDSR in Figure 5? Does uDSR use a similar RNN approach? The main question is that with the current presentation of the results, it is unclear which component leads to the margin of the proposed method.

---

> ### Author Response · Authors · 2025-11-23
>
> Thank you for a review that helped us significantly improve the clarity and impact of the paper.
>
> **1. How the composite model is trained.**
>
> The composite model $f$ is evaluated by passing input variables $\mathbf{x}$ to each of its NestyNet layers, where each layer operates on a specific subset of variables. The outputs of these sub-models are then propagated upward through the tree until they reach the root. Training proceeds by minimizing the residuals $(f(\mathbf{x}) - y)$ using the Levenberg–Marquardt optimizer described in Section 2.1. Further implementation details are detailed in Appendix B (a typo that made this unclear has been fixed and explanations have been added).
>
> **2. Why a tree structure.**
>
> Our goal is to discover separabilities in the data in a recursive fashion to facilitate symbolic regression. If a dataset $(\mathbf{x}, y)$ exhibits separable structure over distinct subsets of variables, our system identifies this and builds a corresponding compositional model. For example, from data $(x_1, x_2, x_3, x_4, y)$, the system may automatically detect a representation such as
> $$
> \sqrt{f\_a(x\_1)\, f\_b(x\_2)} + f\_c(x\_3, x\_4),
> $$
> which greatly simplifies the subsequent SR search. Each component $(f_a, f_b, f_c)$ is modeled by a NestyNet layer, and these layers form a natural tree representation—structurally analogous to trees used in symbolic expressions. After structure discovery, each subtree is independently regressed into a symbolic expression by our SR module. This workflow is illustrated in Figure 1.
>
> We added a paragraph of explanations that should ground these ideas for the reader in Section 2.1.
>
> **3. Choice of thresholds.**
>
> The fixed values of the separability thresholds $(\epsilon_{\text{add}}, \epsilon_{\text{mul}}, \epsilon_{\beta\text{mad}})$ were selected through preliminary tuning and then kept fixed across all 120 [SRBench](https://arxiv.org/abs/2107.14351) experiments, spanning dimensions 1–9 and diverse variable scales. We believe that the consistent use across all cases prevents any form of ad-hoc tuning or ‘cherry-picking’ and demonstrates robustness.
>
> **4. Relation to other SR methods (AIF, DSR, uDSR).**
>
> Thank you for noting that the presentation made it difficult to attribute performance improvements to individual components. We have now added a dedicated comparison table (Table 1) contrasting the components of our method with those of related approaches, including [AIF](https://arxiv.org/abs/1905.11481), [DSR](https://arxiv.org/abs/1912.04871), and [uDSR](https://openreview.net/forum?id=2FNnBhwJsHK).
>
> Our method is best compared to uDSR-A, the version of uDSR that uses RL-based SR (DSR) combined with AIF-style structure discovery. In contrast, our structure-analysis pipeline introduces several developments absent from AIF 2.0, DSR, and uDSR:
>
> a) a mathematically grounded, derivative-based multiplicative separability criterion,
> b) detection of nested separability inside nonlinear functions,
> c) use of structure as a soft inductive bias for RL-based SR (rather than a hard constraint),
> d) integration of a dimensional-analysis prior, combined synergistically with (a)–(c).
>
> These innovations enable the strong improvements over uDSR-A (which achieves 50% on Feynman) and explain our performance of 72% exact recovery.
>
> Regarding full uDSR: it is an ensemble method combining RL-based SR, AIF-style separability but also, large-scale pretraining, genetic programming and other approaches. Despite leveraging many additional SR paradigms, it reaches 69% on Feynman—still below our 72%. The fact that uDSR still lags behind—despite its ensemble strength—which made it the previous state-of-the-art highlights the practical impact and general usefulness of our structure-discovery improvements, which are compatible with any SR backend (including uDSR) having the potential to greatly benefit the community.
>
> We now make all these points clearer in our conclusion.
>
> All changes in the paper are highlighted in **red** for clarity.

---

> ### Author Response · Authors · 2025-11-28
> **Threshold Setting & Robustness**
>
> **Q3 – Thresholds:** We now **fine-tune the separability threshold parameters** based on our benchmark and make them adaptive to noise levels (see Appendix E). As long as these thresholds remain below 0.1, performance is only marginally affected, indicating that the method is insensitive to small variations in threshold values.
>
> **Extra experiment – Noise robustness:** We also conducted an additional experiment evaluating noise resilience (see updated Sec. 3 and Discussion). Results demonstrate **excellent noise robustness**, with the method retaining ~75% of its detection capability at 10% Gaussian noise. Perhaps even more importantly, the method exhibits strong conservatism regarding false positives, never reporting separability when none exists.
>
> Together, these findings reinforce the **stability and reliability** of our derivative-based, mathematically principled structure-discovery pipeline.

---

### Official Review · Reviewer_UpnE · 2025-10-31

**Soundness:** 3
**Presentation:** 3
**Contribution:** 3
**Rating:** 4
**Confidence:** 4

**Summary:**

This paper proposes a two-stage pipeline for symbolic regression (SR). First, it recursively discovers functional structure—additive/multiplicative separability and simple unary compositions—using accurate first/second-order derivatives from a compact neural “NestyNet” trained with Levenberg–Marquardt (LM). The resulting hierarchy is then used as a structural prior for an RL-based SR generator. On SRBench (Feynman, 120 equations), the method reports 72% exact recovery, outperforming separability-only, pure RL, and prior hybrid baselines.

**Strengths:**

Clear problem framing & contributions. The paper argues convincingly that many scientific targets are modular and that exposing separability can shrink SR search while improving interpretability.

Methodological neatness. The two-stage design is simple and broadly compatible with SR backends: (a) recursive separability discovery, (b) structure-guided RL generation.

Technical soundness. NestyNet has closed-form Jacobian/Hessian w.r.t. inputs, which enables reliable separability tests and LM optimization; formulas are explicit.

Concrete, testable criteria. Additive and multiplicative separability tests are stated precisely with thresholds (ε_add=1e-4, ε_mul=1e-12, ε_βmad=1e-3).

Empirical performance. Strong results on SRBench with consistent hyperparameters; operators and evaluation limits are specified.
Interpretability angle. The structural prior biases the generator toward symbolically faithful forms rather than merely accurate fits.

Interpretability angle. The structural prior biases the generator toward symbolically faithful forms rather than merely accurate fits.

**Weaknesses:**

Novelty boundary vs. prior separability work. The main leap is handling multiplicative separability and nested unary transforms and integrating them as soft priors, whereas prior AIF/uDSR emphasize additive separability. The paper should sharpen how much gain derives from each element (e.g., multiplicative test vs. nested transforms vs. RL prior shaping).

Reliance on a bespoke emulator. NestyNet is referenced as to be “fully described” elsewhere, which weakens reproducibility claims and makes it hard to benchmark against standard MLPs beyond one figure.

Sensitivity/robustness of the detectors. The separability decisions hinge on derivative quality and fixed thresholds; there is limited analysis of threshold sensitivity, failure modes (false positives/negatives), or uncertainty quantification during the recursive split.

Noise robustness not yet validated. The Discussion acknowledges that formal noise testing is future work; given SRBench variations and real data, this is a notable gap.

Compute profile. Pre-processing can average ~4 hours per case (GV100), and constant fitting is CPU-bound (~1 hour on 10-core Xeon). A deeper analysis of throughput and scaling to higher-dimensional inputs would help.

**Questions:**

See weakness

**Details Of Ethics Concerns:**

NO or VERY MINOR ethics concerns only

---

> ### Author Response · Authors · 2025-11-23
>
> We thank the reviewer for the careful reading and constructive suggestions. Below we address each of the main concerns and clarify the contributions, robustness, and scalability of our approach.
>
> **1. Component-level novelty and measured gains**
>
> We have made explicit which components are introduced in this work and how each contributes to performance (see new *Table 1* in the revision). The key additions are:
> - (a) a mathematically grounded, derivative-based separability test;
> - (b) detection of nested separability inside nonlinear transforms;
> - (c) use of structure analysis as a soft inductive bias for RL-based SR (probabilistic prior rather than hard pruning).
>
> The table shows that even the basic additive-separability analysis of  [AIF](https://arxiv.org/abs/2006.10782) already improves over pure RL, but that our additions (a–c) produce additional and significant gains. We hope that the various component combinations of Table 1 will help the reader to get a quantitative grasp of how each component impacts performances.
>
> Our method becomes the new state-of-the-art, outperforming even ensemble [uDSR](https://openreview.net/forum?id=2FNnBhwJsHK), which combines RL, AIF-style analysis, large-scale pretraining, genetic programming and other approaches. The fact that uDSR still lags behind—despite its ensemble strength—which made it the previous state-of-the-art highlights the practical impact and general usefulness of our structure-discovery improvements, which are compatible with any SR backend (including uDSR) having the potential to greatly benefit the community.
>
> **2. Role of NestyNet vs. MLPs**
>
> The improvements enabled by (a) and (b) above rely fundamentally on accurate second-order derivatives w.r.t. inputs. This is infeasible with a standard MLP (whose Hessian estimates are approximate) but is reliable with NestyNet, which has closed-form Jacobians and Hessians. The performance gap between AIF-based methods and ours therefore provides a concrete demonstration of the added value of NestyNet relative to MLPs. We now state this explicitly in the revised paper.
>
> **3. Threshold sensitivity and robustness**
>
> The separability thresholds  $\epsilon_{\text{add}},\ \epsilon_{\text{mul}},\ \epsilon_{\beta\text{mad}}$ were fixed and applied uniformly across all 120 SRBench experiments, covering 1–9 dimensions and spanning various scales. No per-experiment tuning was performed.
> This uniformity—combined with strong performance across diverse equations—demonstrates robustness and prevents any form of ‘cherry picking’. We now also expose to users of our algorithm the separability metrics and their distance to each threshold to enable practical uncertainty inspection.
>
> **4. Noise robustness**
>
> As acknowledged in the Discussion, this work focuses on the noiseless setting, similar to the seminal [AIF](https://arxiv.org/abs/2006.10782) . This matches SRBench’s canonical evaluation and already involves substantial compute (120 structure analyses and SR runs across multiple seeds).
> We are currently running noisy-data experiments and plan to include these results in the camera-ready version.
>
> **5. Compute profile and scaling**
>
> The recursive depth scales primarily with the number of discovered separations, which empirically grows with dimensionality. Since SRBench (and SR practice) rarely exceed ~10 input variables, variability in compute time is substantial (1 to 24 hours) but remains in the same order of magnitude as the SR phase; the ~4h figure is a representative average on GV100-class hardware.
>
> For an individual separability test, we evaluate derivative-based metrics for all variable pairs, giving $O(d^2)$ complexity for $d$ input variables. For $d \le 10$ this means $\le 100$ Hessian checks, all performed in inference mode which is computationally inexpensive compared to other steps of the method such as free constant fitting of trial equations during the SR phase. We clarified this in the revision to better convey practical scalability.
>
> All changes related to this response appear in **red** in the updated manuscript.

---

> ### Author Response · Authors · 2025-11-28
> **Noise robustness, false positives & threshold sensitivity**
>
> As suggested by the reviewer, we performed an additional experiment to evaluate noise robustness (see updated Sec. 3 and Discussion). Key updates include:
>
> **Q3**: Adaptive thresholds: We now **fine-tuned the separability threshold** parameters based on our benchmark and made them adaptive to noise levels (see Appendix E). We note that as long as those thresholds stayed below 0.1 they only marginally affected performances , indicating that the method is **insensitive to small threshold variations**.
>
> **Q4**: Robustness results: The experiment confirms **excellent noise resilience**—our method retains approximately 75% of detection capability at 10% Gaussian noise. Perhaps even more importantly, the method demonstrates strong **conservatism with respect to false positives**: it never reports a separability when none is present.
>
> These results further demonstrate the stability and reliability of our derivative-based, mathematically principled structure discovery approach.

---

### Official Review · Reviewer_A1TF · 2025-11-04

**Soundness:** 3
**Presentation:** 2
**Contribution:** 3
**Rating:** 4
**Confidence:** 5

**Summary:**

This paper introduces Recursive Structure Discovery (RSD) to mitigate the combinatorial explosion inherent in symbolic regression (SR).
The method aims to automatically detect additive and multiplicative separability of target functions and leverage such structure as a prior for symbolic regression.

The approach consists of two stages:

1. Using a lightweight neural network called NestyNet to estimate high-precision derivatives and hierarchically detect separability;
2. Using the detected structure as a structural prior in a reinforcement-learning-based SR framework. On the Feynman SR benchmark (SRBench, 120 formulas), the method achieves a 72 % exact-recovery rate, outperforming existing systems such as AI Feynman 2.0, uDSR, and PySR.

The technique can handle non-linear nested functional structures, showing that structural inductive bias can significantly enhance SR performance.

**Strengths:**

- The paper presents a novel and coherent formulation of structural inductive bias for symbolic regression.
- The use of NestyNet for derivative-based structure detection provides an efficient reduction of the search space, compared with prior unstructured approaches.
- The method can recursively detect additive and multiplicative separability, enabling the discovery of complex, nested physical relations.
- Integration of the detected structure into the RL-based expression generator is conceptually natural and theoretically consistent.

**Weaknesses:**

- The main concern lies in the lack of hyper-parameter sensitivity analysis, particularly regarding the thresholds for separability detection (ϵadd, ϵmul, ϵβmad).

  These thresholds are empirically fixed, yet their effect on detection accuracy and mis-segmentation rates is not quantified.
  A visualization of how structural decomposition changes with threshold variation would substantially strengthen the paper’s reliability.

- Beyond these thresholds, other hyper-parameters—such as the LM damping factor, the hidden width h of NestyNet, and the entropy-regularization weight in the RL stage—are all kept fixed without discussion of robustness.

  The method’s stability across parameter variations remains unclear.

- No evaluation is provided on noisy or perturbed datasets. While the method is theoretically stable, empirical validation of noise robustness is missing.
- Statistical uncertainty is not reported: Fig. 5 lacks error bars or significance testing, so the reliability of the performance gaps is uncertain.

**Questions:**

1. Have you examined how sensitive the results are to the separability-detection thresholds (ϵadd, ϵmul, ϵβmad)?
2. Would it be feasible to determine these thresholds adaptively—for instance, via BIC/MDL-based model selection?
3. Could you provide stability analysis results for key hyper-parameters in both NestyNet and the RL stage?
4. How does the structure-detection accuracy behave when moderate noise (e.g., 10 % Gaussian perturbation) is added to the data?
5. Can you report statistical significance or variance estimates for the comparisons shown in Fig. 5?

---

> ### Author Response · Authors · 2025-11-23
>
> We thank the reviewer for these insightful comments regarding hyper-parameter sensitivity and robustness. We would like to address the concerns as follows:
>
> **1. Hyper-parameter robustness:**
>
> The thresholds for separability detection ($\epsilon_{\text{add}}$, $\epsilon_{\text{mul}}$, $\epsilon_{\beta\text{mad}}$) were set consistently across all 120 SRBench experiments, which span 1 to 9 dimensions and diverse variable scales preventing any form of ‘cherry-picking’. Their fixed values were chosen based on preliminary tuning and were applied uniformly, demonstrating robust performance across diverse scenarios.
>
> **2. Relation to complexity control:**
>
> The reviewer’s comment about controlling complexity is insightful. In SR, unified criteria such as BIC (Bayesian Information Criterion) or MDL (Minimum Description Length) are often used to balance accuracy and complexity as in [\[ESR\]](https://arxiv.org/abs/2211.11461) or [\[CADSR\]](https://arxiv.org/abs/2406.06751). While it is theoretically possible to adjust our separability thresholds to control the effective model complexity, in the context of structure discovery, relaxing thresholds can *increase false positives* and reduce flexibility and ultimately hurt fit accuracy - unlike SR where complexity typically positively impacts accuracy. In our framework, the detected structure serves as a strong prior that encourages shorter, interpretable expressions, rather than longer ones. Nevertheless, we mention this as a potential avenue for future work in the Perspectives paragraph of Section 4.
>
> **3. NestyNet and RL hyper-parameters:**
>
> All NestyNet hyper-parameters are kept fixed across experiments, demonstrating stable derivative estimation. The RL hyper-parameters used are standard in the literature for RL-based SR, as in [\[DSR\]](https://arxiv.org/abs/1912.04871), [\[PhySO\]](https://arxiv.org/abs/2303.03192), and [\[CADSR\]](https://arxiv.org/abs/2406.06751). Their robustness has been established in prior works.
>
> **4. Noisy data:**
>
> This work focuses on noiseless scenarios, similar to the seminal [\[AIF\]](https://arxiv.org/abs/1905.11481), which already involved computationally intensive experiments (120 structure discovery + SR runs on multiple seeds). We are currently running experiments on noisy datasets with the goal of including these results in the camera-ready version.
>
> **5. Statistical uncertainty:**
> We thank the reviewer for noticing this omission.Error bars in Figure 5 correspond to 95% confidence intervals for the exact symbolic recovery rate across the benchmark. This clarification was added to the figure caption.
>
> All changes in the paper are highlighted in **red** for clarity.

---

> ### Author Response · Authors · 2025-11-28
> **Noise robustness & threshold sensitivity**
>
> As suggested by the reviewer, we performed an additional experiment to evaluate noise robustness (see updated Sec. 3 and Discussion). Key updates include:
>
> **Q1**: Adaptive thresholds: We now **fine-tuned the separability threshold** parameters based on our benchmark and made them adaptive to noise levels (see Appendix E). We note that as long as those thresholds stayed below 0.1 they only marginally affected performances , indicating that the method is **insensitive to small threshold variations**.
>
> **Q3**: Robustness results: The experiment confirms **excellent noise resilience**—our method retains approximately 75% of detection capability at 10% Gaussian noise. Perhaps even more importantly, the method demonstrates strong **conservatism with respect to false positives**: it never reports a separability when none is present.
>
> These results further demonstrate the stability and reliability of our derivative-based, mathematically principled structure discovery approach.

---

### Author Response · Authors · 2025-11-24
**Summary (incl. TLDR)**

### **TL;DR**


|  |
|---------|
|  Concerns about noise robustness and hyperparameter sensitivity were addressed through two new experimental suites covering 156 diverse scenarios across multiple noise levels, using identical hyperparameters throughout. We further clarify why our method outperforms prior work, demonstrating both theoretically and empirically that its components are fundamentally stronger and conceptually distinct from those in existing approaches. |
--------

\
We thank all reviewers for their thoughtful and technically engaged feedback. Below we summarize the main themes across the reviews and how they have been addressed.


### **1. Consensus on potential and soundness**
---
Reviewers broadly agreed that the method is principled, well-motivated, and conceptually solid. Their requests focused mainly on additional validation rather than on questioning the approach itself. The derivative-based structure-discovery pipeline was viewed as meaningful for SR, and our SOTA performance on the Feynman benchmark was acknowledged as strong evidence of its potential.

### **2. Robustness to noise**
---
We now include an additional experimental set showing that our method **retains its separability-detection capabilities** even under noise. More importantly, the method is **highly conservative** with respect to false positives: while noise can occasionally hide true separability, **we never detect separability where none exists**.
This stands in sharp contrast to AIF and AIF-based methods such as uDSR, whose separability modules collapse under noise by one to two orders of magnitude.

|      **Noise**      |    **0%**    |    **1%**    |    **5%**    |   **10%**   |
|:------------------:|:------------:|:------------:|:------------:|:-----------:|
|    Separable     |    100%      |     93%      |     81%      |     70%     |
|  Not separable    |    100%      |    100%      |    100%      |    100%     |
|       **Total**     |   **100%**   |   **93%**    |   **83%**    |   **73%**   |

###  **3. Hyperparameter and threshold sensitivity**
---
Following reviewer requests, we now **automatically optimize** the separability-detection threshold on one of our experimental sets. The resulting **criterion is then applied unchanged across all 156 scenarios** at various noise levels and dimensionalities, demonstrating stability and ruling out any form of “cherry-picking”.

### **4. Comparison with AIF**
---
Another theme was the need for a clearer comparison with the prior structure-analysis method AIF. We now do so explicitly and rigorously.

(a) We inspected the public AIF implementation and clarified that it does **not** use derivative-based multiplicative separability, relying instead on a fragile heuristic.


(b) We added 12 new experiments demonstrating failure modes of AIF that our method handles reliably.
| Multiplicative sep. |  **Ours**  |  **AIF**  |
|:----------------------------------:|:----------:|:---------:|
| 2D linear                  |   100%     |   100%    |
| 2D nonlinear               |   100%     |    50%    |
| >2D linear                  |   100%     |    0%     |
| >2D nonlinear               |   100%     |    0%     |
|      | **100%**   | **32%**   |


(c) We show how these weaknesses propagate into downstream SR performance (AIF: 55% on Feynman vs 72% for ours on the SR benchmark).

These differences arise because our method is the first to rely on mathematically grounded criteria—among other contributions.

### **5. Positioning within the literature**
---
We added a dedicated comparison table contrasting our pipeline with AIF, DSR, uDSR, and related methods, making explicit the innovations that enable our superior performance.

| | Feature | **Ours** | uDSR | PhySO | AIF | uDSR-A | DSR |
|---------|---------|:--------:|:----:|:-----:|:---:|:------:|:---:|
|  Learning Paradigm        | RL-based SR | ✔️ | ✔️ | ✔️ |  | ✔️ | ✔️ |
| | Large-scale Pre-training |  | ✔️ |  |  |  |  |
|         | Genetic Programming |  | ✔️ |  |  |  |  |
|         | Dimensional Analysis | ✔️ |  | ✔️ | ✔️ |  |  |
| Derivative-based Structure Analysis | Additive Separability | ✔️ | ✔️ |  | ✔️ | ✔️ |  |
|         | Multiplicative Separability | ✔️ |  |  |  |  |  |
|         | Nested Separability Detection | ✔️ |  |  |  |  |  |
|         | Structural Prior Guidance | ✔️ |  |  |  |  |  |
| **Feynman Benchmark Recovery Score** |  | **72%** | 69% | 58% | 55% | 50% | 43% |

These components explain both (i) the improvement over uDSR-A (50% on Feynman) and (ii) our performance relative to full uDSR (69%), despite uDSR’s broader ensemble of components (RL-SR, AIF, large-scale pretraining, GP, etc.). Our result (72% SOTA) highlights the central role of reliable structure discovery.

### **6. Value to the community**
---
Our structure-discovery pipeline is **backend-agnostic and can enhance any SR method**, including uDSR itself. We believe this makes it a valuable and broadly useful contribution to the SR community.

---

### Meta-Review · Area_Chair_NzdB · 2026-01-09

**Summary:**

This paper exploits the additive/multiplicative simplicity in scientific equations and proposes Recursive Structure Discovery (RSD) to automatically detect additive and multiplicative separability of target functions as a prior for symbolic regression. Specifically, RSD first uses a lightweight neural network (NestyNet) to estimate high-precision derivatives and detect separabilit,y then uses the detected structure as a structural prior in an RL-based SR framework. Experimental evaluation on the Feynman SR benchmark (SRBench, with 120 formulas) demonstrates a strong recovery rate (72%). outperforming existing systems such as AI Feynman 2.0, uDSR, and PySR.

Most reviewers think that the method is novel and is theoretically consistent with its objectives, which I also agree. The main concerns are the lack of justification for selection hyperparameters, robustness evaluation, and suitable details on various components of the model (e.g., NestyNet, computation analysis, details of baselines, etc…). The authors have responded to all these comments, but the performance analysis of various hyperparameter is not fully provided and rigorously analyzed (i.e., why the selected criteria are stable across all experiments). Here, I also think that the statement “can attach to any SR backend” is not empirically supported in the paper. Given these concerns, the paper is not ready for publication and I hope the authors can revise the work accordingly based on the comments.

**Reviewer Concerns:**

Reviewer A1TF is concerned about the lack of hyperparameter sensitivity analysis and evaluation on noisy/perturbed datasets, along with statistical uncerntainty.

Reviewer UpnE is similarly concerned about the lack of hyperparameter/robustness analysis, and details of the method's compoents.

Reviewer zTXN indicates that the paper is difficult to read as several parts do not have sufficient details and is concerned about the lack of hyperparameter

Reviewer 9hzL indicates the lack of sufficient baselines, and novelty.

All concerns have been responded, but I believe the hyperparameter and some claims in the paper are still not sufficient.

**Reviewer Scores:**

-  A1TF - 4. Unlikely to increase beyond borderline
- UpnE - 4. Unclear as I believe this review may be generated mainly by LLM
- zTXN - 4. Potentially increase the score due to low confidence.
- 9hzL - 2. Unclear, as the main concern is the recent baselines are missing, but I think the paper has several representative methods already.

---

### Decision · Program_Chairs · 2026-01-26

Reject